

# Dissolved organic matter release by phytoplankton in the context of the Dynamic Energy Budget theory

Eleni Livanou[1,2], Anna Lagaria[2], Stella Psarra[2], and Konstadia Lika[1]

[1]Department of Biology, University of Crete, 70013 Heraklion, Greece
[2]Hellenic Centre for Marine Research, Institute of Oceanography, 71500 Heraklion, Greece

*Correspondence to:* K. Lika (lika@uoc.gr)

**Abstract.** Extracellular release of dissolved organic matter (DOM) by phytoplankton is a significant process that drives the microbial loop, providing energy and nutrients to bacteria. In this paper, a dynamic energy budget model is proposed for describing DOM release by phytoplankton under nitrogen and phosphorus limiting conditions. The model allows for the distinction of the two major mechanisms of DOM release; passive diffusion related to growth and lysis of the cells and active exudation related to rejection of unprocessed substrates due to stoichiometric constraints. Model results suggest that phosphorus deficiency has less severe effect on phytoplankton growth and primary production (PP) rate than nitrogen deficiency, while co-limitation by both nutrients has the most severe effect. The dependence of dissolved organic carbon (DOC) release rate on the cellular carbohydrates concentration is also highlighted by the model. Furthermore, model predictions resolve the relationship between PP and DOC release under different nutrient availability scenarios, providing a possible explanation for the deviations from 1:1 linear relationship between PP and DOC release, often observed in oligotrophic systems. This deviation is a result of the prevalence of the active exudation mechanism and the reduction of the PP rate due to nutrient limitation. Conversely, passive diffusion is more important under nutrient-replete conditions. The different relative contributions of the two mechanisms result in different qualities of DOM produced by phytoplankton in terms of elemental and molecular composition and size fractions, with potential implications for the bioavailability of the produced DOM for bacteria and the coupling of phytoplankton–bacteria dynamics.

## 1 Introduction

More than half of the organic carbon fixed by phytoplankton ends up in the dissolved organic matter (DOM) pool, constituting a major carbon source for heterotrophic procaryotes (del Giorgio and Cole, 1998; Ducklow and Carlson, 1992). The release of organic matter from phytoplankton cells is mediated by various processes such as grazing and sloppy feeding (Møller, 2007), viral lysis (Fuhrman, 1999), cell death and lysis (Orellana et al., 2013), as well as through active and/or passive release from healthy cells (Fogg, 1983; Bjørnsen, 1988).

To date, two conceptual processes have been proposed to describe the extracellular release of DOM by phytoplankton: the passive diffusion model and the overflow model. The passive diffusion model refers to the leakage of low molecular weight compounds through the cell membrane (Bjørnsen, 1988). This mode of release takes place throughout all stages of growth



and correlates with phytoplankton cell size, while, according to this mechanism, the relative contribution of smaller cells to extracellular release of DOM is expected to be higher due to their high surface-to-volume ratio (Bjørnsen, 1988; Borchard and Engel, 2015). The overflow model describes a physiological mechanism of active excretion of high molecular weight compounds (Fogg, 1983; Borchard and Engel, 2015). It has been suggested that this mechanism takes place under sub-optimal

growth conditions such as under N- and/or P-limitation (Obernosterer and Herndl, 1995; Lagaria et al., 2013) and/or high irradiance (Cherrier et al., 2014), when there is adequate light for photosynthesis and, thus, carbohydrates production, but limited nutrient availability for biosynthesis of the structural components of the cell (Fogg, 1983; Thornton, 2014). Findings from numerous field and culture studies have supported the existence of either one or both conceptual processes of DOM release (Teira et al., 2001b; Chin et al., 2004; Lagaria et al., 2013; López-Sandoval et al., 2013; Myklestad, 2000). However, it

is very difficult to elucidate the underlying cell mechanisms based solely on the empirical observations of these studies, as for example, correlations of DOM release with primary production (PP).

Phytoplankton-derived DOM mostly comprises different forms of carbohydrates depending on the species, the growth status of the cells and nutrient availability (Myklestad, 2000; Underwood et al., 2004; Urbani et al., 2005). It has been suggested that the smaller (<10kDa) size fraction carbohydrates, rich in glucose, pass through the membrane passively and larger and

more complex heteropolysaccharides that contain a variety of monomers (e.g. galactose, mannose, xylose, rhamnose, fucose and arabinose) are exported by active exudation while both processes can take place simultaneously in the cell (Borchard and Engel, 2015). Besides carbohydrates, proteins and free amino acids also constitute a significant part of DOM released by phytoplankton (Granum et al., 2002). Under N-replete conditions, a substantial fraction of the assimilated nitrogen may be excreted as dissolved organic nitrogen (DON) (Nagao and Miyazaki, 2002). Under P-replete conditions, phytoplankton can

produce dissolved organic phosphorus (DOP) (Saad et al., 2016).

The production of DOM by phytoplankton and the subsequent consumption by bacteria are very important processes for the carbon flux in the marine ecosystem. Phytoplankton-derived DOM degradation and bacterial production are affected by the nutrient availability (Puddu et al., 2003; Fouilland et al., 2014). Polysaccharides rich in glucose, produced under nutrient-replete conditions, are easily consumed by bacteria while heteropolysaccharides produced mainly under nutrient-limiting conditions,

are more resistant to bacterial degradation and can, thus, be accumulated in the water (Obernosterer and Herndl, 1995; Urbani et al., 2005; Thingstad et al., 1997; Hama and Yanagi, 2001). Therefore, linking phytoplankton physiology with the composition and subsequent bacterial utilization of photosynthetically-derived DOM is an important step in understanding the patterns of DOM fluxes in the ocean and the relationships between phytoplankton and bacteria.

Due to the central role of phytoplankton-derived DOM for the carbon flux in marine ecosystems, the process of DOM release

is included in many phytoplankton physiological models. DOM release rate is modelled as a percentage of primary production and/or proportional to phytoplankton biomass (Spitz et al., 2001; Keller and Hood, 2011; Hasumi and Nagata, 2014), nutrient limitation (Anderson and Williams, 1998) and cell size (Kriest and Oschlies, 2007). Other models relate DOM release to the nutrient status of the cell, using cell quota or elemental ratios (Baretta-Bekker et al., 1997; Van Den Meersche et al., 2004; Baklouti et al., 2006; Vichi et al., 2007; Schartau et al., 2007; Flynn et al., 2008; Kreus et al., 2014) and, thus, allowing

DOM release rate to dynamically vary with the physiological state of cells. These variations of cell quota are mainly due to



variations of the the stored cellular nutrients and carbohydrates (Geider and La Roche, 2002). Nonetheless, with the exception of the model of Lorena et al. (2010), no attempt has been made to model DOM exudation by phytoplankton in relation to the dynamics of internal reserves. In addition, the majority of phytoplankton physiology models usually refer to N-limited growth, as this represents the dominant condition in the oligotrophic environments of the world ocean (Falkowski et al., 1998), and do

not take into account the effects of concurrent limitation by nitrogen and phosphorus, which in some cases significantly affects DOM release (Lagaria et al., 2011).

In this study, we propose a model for phytoplankton growth and DOM release under nutrient limiting conditions, based on the Dynamic Energy Budget (DEB) theory (Kooijman, 2010). DEB theory is based on physicochemical first principles and, as such, implies mechanistic rules for energy uptake and use by the organism, that apply to all organisms (Sousa et al., 2010).

Responding to the need for models that would incorporate the dynamic cellular functioning (Glibert et al., 2013), the model includes explicitly the uptake, use and excretion of carbon, nitrogen and phosphorus by the individual cell. Thus, the aims of this study are to: 1) investigate the effects of N- and P-limitation on DOM release by phytoplankton, 2) resolve the relationship between primary production and DOM release and 3) elucidate the mechanisms under the two conceptual processes of DOM production, namely, passive diffusion and active exudation.

## 2 Methods

### 2.1 Model overview

The phytoplankton model is based on the modelling framework provided by the DEB theory (Kooijman, 2010). A schematic presentation of the metabolic processes in an individual cell is given in Fig. 1. The model considers four inorganic nutrients; ammonium (NH), nitrate (NO), phosphorus (P) and inorganic carbon (IC), each taken up independently and with an inhibitory

mechanism of NO uptake by assimilated NH (Flynn et al., 1997; Glibert et al., 2015). According to DEB theory, biomass is partitioned into structural mass $M_V$ and reserve mass $M_i$ ($i$ = E, $E_{NH}$, $E_{NO}$, $E_P$, $E_{CH}$), both with constant chemical composition, an assumption called "strong homoeostasis". Elemental composition of biomass is monitored in terms of Carbon (C), Nitrogen (N) and Phosphorus (P). Although both reserves and structure have constant stoichiometry, the proportion of an element in the total biomass may vary due to variation in the relative amount of reserves and structure.

All transformations of substrates are performed using the concept of synthesizing unit (SU) introduced by Kooijman (1998, 2010) and successfully used for modelling the uptake of substrates and synthesis of products in phytoplankton (Papadakis et al., 2005; Lika and Papadakis, 2009; Lorena et al., 2010). The SUs are generalized enzymes that stoichiometrically combine the substrate fluxes and give the product. Photosynthesis light and dark reactions are modelled explicitly, while phytoplankton growth is controlled by the availability of nitrogen and phosphorus. Photosystems harvest light (I) and produce NADPH

(NA), that, along with IC, will produce carbohydrates (CH). Then, the four substrates, NH, NO, P and CH, each with its corresponding assimilation flux, are combined together by the $SU_1$ (Fig. 1) to form the generalized reserves (E) which have a fixed stoichiometry. The reserves are considered as intermediate steps to structure formation. SU dynamics imply some rejected fluxes of the substrates due to stoichiometric constraints. Each rejected substrate "molecule" from $SU_1$ (Fig. 1) is channelled to



the corresponding reserve ($E_{NH}, E_{NO}, E_P, E_{CH}$). A second synthesis ($SU_2$) occurs from the mobilized reserves and the rejected molecules from $SU_2$ are either directed back to the reserves or exported outside the cell. Mobilized reserves are combined to increase structure (V) and pay the maintenance costs (M) of the cell, with maintenance taking priority over growth (G). In addition to the rejected reserve "molecules", that are excreted either in inorganic or organic form, release of organic matter by

5   phytoplankton is also associated with growth (G) and death (D).

The metabolic processes that are schematically presented above and fully described in the following subsections concern an individual cell. In DEB theory, assimilation is considered proportional to the surface area while maintenance is considered proportional to volume of structural mass. However, in the context of DEB theory, unicellular organisms that propagate through division, such as phytoplankton cells, can be considered V1-morphs which implies that surface area is proportional to volume

10   and therefore the whole population can be described by state variables that are the sum of the state variables of each individual cell (Kooijman, 2010; Lorena et al., 2010; Marques et al., 2014). This simplification is possible due to the narrow range of size during cell growth, which implies that the deviations from V1-morphy are irrelevant when investigating population growth. (Kooijman, 2010; Lorena et al., 2010).

The proposed population model has six state variables for the organism (structure,V, ammonium, $E_{NH}$, nitrate, $E_{NO}$, phos-

15   phate, $E_P$, carbohydrates, $E_{CH}$ and generalized, E, reserves) and four for the environment (inorganic carbon, IC, ammonium, NH, nitrate, NO, phosphorus, P) (light is assumed to be constant). In addition, three more variables are monitored to account for the excreted dissolved organic matter (dissolved organic carbon (DOC), dissolved organic nitrogen (DON), and dissolved organic phosphorous (DOP)). The rates of change in the state variables are given by the following differential equations

*Environment-related state variables*

$$\frac{dX_{IC}}{dt} = (-j_{CH,A} + j_{IC,E})X_V$$
$$\frac{dX_{NH}}{dt} = (-j_{NH,A} + j_{NH,E})X_V + (1 - \beta_{DON})hX_{E_{NH}}$$
$$\frac{dX_{NO}}{dt} = (-j_{NO,A} + j_{NO,E})X_V + hX_{E_{NO}}$$
$$\frac{dX_P}{dt} = (-j_{P,A} + j_{P,E})X_V + hX_{E_P} \tag{1}$$





*The organism-related state variables*

$$\frac{dX_E}{dt} = (j_{E,A} - j_{E,C})X_V - hX_E$$

$$\frac{dX_{E_{CH}}}{dt} = (j_{E_{CH},A} - j_{E_{CH},C} + \kappa_{E_{CH}}j_{E_{CH},R})X_V - hX_{E_{CH}}$$

$$\frac{dX_{E_{NH}}}{dt} = (j_{E_{NH},A} - j_{E_{NH},C} + \kappa_{E_{NH}}j_{E_{NH},R})X_V - hX_{E_{NH}}$$

$$\frac{dX_{E_{NO}}}{dt} = (j_{E_{NO},A} - j_{E_{NO},C} + \kappa_{E_{NO}}j_{E_{NO},R})X_V - hX_{E_{NO}}$$

$$\frac{dX_{E_P}}{dt} = (j_{E_P,A} - j_{E_P,C} + \kappa_{E_P}j_{E_P,R})X_V - hX_{E_P}$$

$$\frac{dX_V}{dt} = (r - h)X_V \tag{2}$$

*Variables related to excretion processes*

$$\frac{d}{dt}X_{DOC} = j_{DOC,E}X_V + h(X_E + X_{E_{CH}})$$

$$\frac{d}{dt}X_{DON} = j_{DON,E}X_V + h(n_{NE}X_E + \beta_{DON}X_{E_{NH}})$$

$$\frac{d}{dt}X_{DOP} = j_{DOP,E} + hn_{PE}X_E \tag{3}$$

where $X_i$ is the concentration of compound $i$ and $j_{i,k}$ is the specific (i.e., per unit of structural mass) flux of compound $i$ associated with process $k$, where $k$ = A (assimilation), R (rejection), E (excretion), or C (catabolism). $r$ denotes the net growth rate and $h$ denotes the death rate. The reserves of a dead cell contribute to the dissolved organic and inorganic pools while its structure contributes to the particulate organic compartment (not shown). Variables and parameters in the above and the following model equations are introduced in Table 1.

## 2.2 Model equations

### 2.2.1 Assimilation rates and photosynthesis

The photosynthetic units (PSUs) of phytoplankton converts light and inorganic carbon into autotrophic assimilate. During the light reactions photons bound to the PSUs provide the excitation energy for the formation of NADPH (Papadakis et al., 2005; Lika and Papadakis, 2009) which is produced at a rate $j_{NA,A}$ given by

$$j_{NA,A} = v_P y_{NA,L} \frac{1}{k_L^{-1} + (\rho_L j_L)^{-1}} \tag{4}$$

where $j_L$ is the specific arrival rate of photons taken proportional to the incident light intensity, according to the equation $j_L = \alpha_L I$, with $a_L$ the specific photons arrival cross section. $k_L$ and $\rho_L$ are, respectively, the handling rate and the photon's binding probability. P-limitation can affect the carbon fixation since phosphorus is essential in major cellular functions, such as nucleic acids, ATP and NADPH synthesis (Geider et al., 1993; Kamalanathan et al., 2016). Here we assume that the low P



availability will decrease the amount of NADPH produced per absorbed photon, and subsequently will affect the dark reactions (Geider et al., 1993). $v_P$ is assumed to be an increasing function of the P density of the cell, from a minimum value $v_{Pmin}$ to a maximum value $v_{Pmax}$. A functional form that can capture the described behaviour of $v_P$ is

$$v_P = v_{P_{max}} - (v_{P_{max}} - v_{P_{min}})e^{-b_P q_P^2} \tag{5}$$

where $q_P = \frac{n_{P,E}M_E + M_{E_P}}{n_{P,V}M_V}$ is the phosphorus content in the reserves relative to structure.

The photochemical efficiency of PSII and the chlorophyll a concentration are reduced under nitrogen starvation (Negi et al., 2016; Kamalanathan et al., 2016) as the machinery of light reactions (chloroplasts) is protein-based and therefore nitrogen-rich (Bonachela et al., 2013). This reduction in photochemical efficiency of PSII is captured by the model through the parameter $\rho_L$ that depends on the the N density of the cell and increases from a minimum value $\rho_{L_{min}}$ to a maximum $\rho_{L_{max}}$. A functional

form for $\rho_L$ is given by

$$\rho_L = \rho_{L_{max}} - (\rho_{L_{max}} - \rho_{L_{min}})e^{-b_N q_N^2} \tag{6}$$

where $q_N = \frac{n_{N,E}M_E + M_{E_{NH}} + M_{E_{NO}}}{n_{N,V}M_V}$ is the nitrogen content in the reserves relative to structure.

The assimilation rates of inorganic carbon, phosphorus and ammonium follow classic Michaelis-Menten kinetics and are given by

$$j_{i,A} = j_{i_{max}} \frac{X_i}{X_i + K_i}, \quad (i = \text{IC, P, NH}) \tag{7}$$

where $j_{i_{max}}$ is the maximum assimilation rate of substrate $i$ and $K_i$ is the half-saturation constant.

The assimilation of nitrate is inhibited by a product of ammonium assimilation (Flynn et al., 1997; Glibert et al., 2015) and is modelled using the binding-inhibition kinetics (Kooijman, 2010). According to this scheme the specific assimilation flux of nitrate is given by

$$j_{NO,A} = \frac{j_1'}{\left(1 + \frac{j_1'}{k_1} + \frac{j_2'}{k_2}\right)} \tag{8}$$

where $j_1' = \rho_1 \alpha_{NO} X_{NO}$ and $j_2' = \rho_2 j_{NH,A}$ are the specific arrival rates of nitrate and ammonium, respectively. $\rho_*$ the binding probabilities and $k_*$ the handling rates (the background for the formulation of Eq. (8) can be found in supplementary information (Sect. S1.)).

In the Calvin–Benson cycle, NADPH and inorganic carbon are complementary substrates processed in parallel by the $\text{SU}_{CB}$

(Fig. 1) to form carbohydrates (Papadakis et al., 2005; Lika and Papadakis, 2009). According to the rules of this transformation (Lika and Papadakis, 2009) the specific CH assimilation flux is given by

$$j_{CH,A} = \left(\frac{1}{k_{CH}} + \frac{1}{j_{IC,A}'} + \frac{1}{j_{NA,A}'} - \frac{1}{j_{IC,A}' + j_{NA,A}'}\right)^{-1} \tag{9}$$

where $j_{i,A}' = \rho_i \frac{j_{i,A}}{y_{i,E_{CH}}}$ is the specific arrival rate of substrate $i$ (IC or NA), $\rho_i$ is the binding probability to the $\text{SU}_{CB}$, $y_{i,E_{CH}}$ is the stoichiometric coefficient that denotes the amount of compound $i$ consumed per amount of $\text{E}_{CH}$ produced and $k_{CH}$ is

the handling rate of the two substrates. As mentioned above, nitrogen and phosphorus content in reserves relative to structure affect NADPH production rate (Eqs. 5, 6) and, thus, the $j_{CH,A}$ flux.





### 2.2.2 *Reserve formation*

Generalized reserves, E, are formed directly from the assimilation fluxes of CH, NH, NO and P, given by Eqs. (7) – (9). In this transformation, using SU dynamics, NH and NO are treated as substitutable substrates and complementary to CH and P and their binding is parallel. In the current model, we assume that phosphorus and ammonium can directly be used by the SU to form the generalized reserves. However, nitrate must be first reduced to nitrite in the cytoplasm and then further reduced to ammonium in the chloroplasts (Glibert et al., 2015). The extra energy requirements of nitrate reduction is taken into account in the model by the stoichiometric coupling of carbohydrates utilization with the nitrogen substrate that is used. According to (Kooijman, 2010) (Eq. 5.23, extended to include phosphorus), the specific assimilation flux of generalized reserves is

$$
\begin{aligned}
j_{E,A} = \Big( (j_{E,A_m})^{-1} &+ j_{CH,A}^{'-1} + (j_{NO,A}^{'} + j_{NH,A}^{'})^{-1} + j_{P,A}^{'-1} \\
&- (j_{CH,A}^{'} + j_{NO,A}^{'} + j_{NH,A}^{'})^{-1} - (j_{CH,A}^{'} + j_{P,A}^{'})^{-1} \\
&- (j_{P,A}^{'} + j_{NO,A}^{'} + j_{NH,A}^{'})^{-1} \\
&+ (j_{CH,A}^{'} + j_{NO,A}^{'} + j_{NH,A}^{'} + j_{P,A}^{'})^{-1} \Big)^{-1}
\end{aligned}
\tag{10}
$$

where $j_{i,A}^{'} = \rho_i j_{i,A}/y_{i,E}$ denotes the specific arrival rate of substrate $i$ (CH, NH, NO, P), $\rho_i$ its binding probability to the SU, and $y_{i,E}$ the yield coefficient that represents the mole of the compound $i$ required to form one mole of the generalized reserves, E. The stoichiometric coefficient $y_{CH,E}$ that couples carbohydrates to the generalized reserves yield is not constant but it depends on the nitrogen source that takes part in the transformation each time, according to the relationship: $y_{CH,E} = \theta_{NH}^A y_{CH,E}^{NH} + \theta_{NO}^A y_{CH,E}^{NO}$, where $\theta_{NH}^A + \theta_{NO}^A = 1$ and $\theta_{NH}^A = j_{NH,A}^{'}(j_{NO,A}^{'} + j_{NH,A}^{'})^{-1}$ and $y_{CH,E}^{NH} < y_{CH,E}^{NO}$. $j_{E,A_m}$ denotes the structure-specific maximum assimilation rate of generalized reserves and also depends on the nitrogen source: $j_{E,A_m} = \theta_{NO}^A/k_{NO} + \theta_{NH}^A/k_{NH}$, with $k_{NO}$, $k_{NH}$ being the handling rates of nitrate and ammonium, respectively.

Since the nutrients are taken up independently, one or more assimilation fluxes can limit the synthesis of the generalized reserves. In that case, the non limiting assimilated "molecules" will occupy the binding sites of the SU but they won't be processed further due to the absence of the limiting flux (Lorena et al., 2010) and, thus, they will be rejected by the SU. The "molecules" of the compound $i$ rejected from the $SU_1$ (Fig. 1) are stored in the corresponding reserves. Nitrate can be accumulated in large quantities in the cell and in the current model it is contained in the $E_{NO}$ reserve (Dortch et al., 1984; Glibert et al., 2015). On the other hand, ammonium is found rarely in large quantities while free amino acids can be accumulated in large quantities in the cell (Dortch et al., 1984; Glibert et al., 2015). Given that the incorporation of ammonium into organic molecules is a fast process (Stolte and Riegman, 1995) we assume that the $E_{NH}$ reserve contains a mixture of ammonium and DON fraction of free amino acids. Accordingly, we assume that the DOC fraction of the free amino acids is contained in the $E_{CH}$ reserves. The specific assimilation fluxes for $E_{CH}$, $E_P$, $E_{NH}$, $E_{NO}$ are given by

$$
\begin{aligned}
j_{E_i,A} &= j_{i,A} - y_{i,E} j_{E,A} && (i = \text{CH,P}) \\
j_{E_i,A} &= j_{i,A} - \theta_i^A y_{N,E} j_{E,A} && (i = \text{NH,NO})
\end{aligned}
\tag{11}
$$





### 2.2.3 *Growth rate*

All reserves are mobilized to allocate energy to growth and maintenance. According to DEB theory, reserve densities, $m_E = M_E/M_V$ and $m_{E_i} = M_{E_i}/M_V$ with $i$ = CH, NH, NO, P follow first-order kinetics, which means that the structure-specific catabolic fluxes $j_{E,C}$ and $j_{E_i,C}$ are given by

$$j_{E,C} = m_E(k_E - r) \quad \text{and} \quad j_{Ei,C} = m_{E_i}(k_{E_i} - r) \tag{12}$$

with $k_E, k_{E_i}$ the reserves turnover rates, which are constant, and $r$ the net specific growth rate, $r = \frac{1}{M_V}\frac{dM_V}{dt}$. The specific catabolic fluxes represent the mobilized reserve per unit of structural mass that will be used for growth and maintenance, with maintenance taking priority over growth.

E$_i$ reserves send their specific catabolic fluxes $j_{Ei,C}$, to the synthesizing unit SU$_2$ (Fig. 1) to form a generalized reserve E′,

in an analogous transformation as for E. The specific catabolic flux, $j_{E',C}$ can be calculated from Eq. (10) by replacing $j_{*,A}$ with $j_{*,C}$ (Kooijman, 2010).

The two resulting catabolic fluxes, $j_{E,C}$ and $j_{E',C}$ are combined together by the growth SU, which is assumed to be fast enough to avoid spoiling of reserves, to form the flux for growth $j_{VG}$

$$j_{VG} = (j_{E,C} + j_{E',C} - j_{E,M})y_{E,V}^{-1} \tag{13}$$

The term $j_{E,M}$ stands for the structure specific maintenance flux and it is assumed to be constant. When the catabolic fluxes $j_{E,C} + j_{E',C}$ are not enough to cover the maintenance requirements, the remainder will be paid from structure at a rate $j_V^M = (j_{E,M} - \min(j_{E,C} + j_{E',C}, j_{E,M})y_{E,V}^{-1}$. The inclusion of the term $j_V^M$ allows the net specific growth rate, $r = j_{VG} - j_V^M$ to be negative.

### 2.2.4 *Release of dissolved organic matter and inorganic nutrients*

In the context of DEB theory, exudation of DOM by phytoplankton can be described when considering the two possible contributing fluxes: rejection flux and product synthesis (Lorena et al., 2010). The catabolic fluxes from the E$_{CH}$, E$_P$, E$_{NH}$, E$_{NO}$ that cannot be used for synthesis will be rejected by the SU$_2$ at rates

$$j_{E_i,R} = j_{Ei,C} - y_{i,E'}j_{E',C} \qquad (i = \text{CH,P})$$
$$j_{E_i,R} = j_{Ei,C} - \theta_i^C y_{i,E'}j_{E',C} \quad (i = \text{NH,NO}) \tag{14}$$

These $j_{E_i,R}$ rejection fluxes are further divided into two types of fluxes: a fixed fraction $\kappa_{E_i}$ that is fed back to the respective reserve and the rest $(1 - \kappa_{E_i})$ that is excreted in the environment (Kooijman, 2010; Marques et al., 2014). This rejection flux directed outside the cell is linked to the active exudation of the non-limiting compounds into the environment (Myklestad, 1995; Fogg, 1983). This flux is dynamic and depends on the nutrient status of the cell. Based on experimental evidence about the partition of the excreted non-limiting compounds (see supplementary information (Sect. S2)), we assume, as in Grossowicz

et al. (2017), that, a fraction $\kappa_{X_i}$ is excreted in the inorganic form, thus

$$j_{i,R} = \kappa_{X_i}(1 - \kappa_{E_i})j_{E_i,R} \quad (i = \text{NH,NO,P}) \tag{15}$$




while a fraction $(1 - \kappa_{X_i})$ is excreted in the organic form of $X_i$ (i.e., DON for $i$= NH, NO, DOP for $i$ = P), thus

$$j_{\ell,R} = (1 - \kappa_{X_i})(1 - \kappa_{E_i})j_{E_i,R} \quad (\ell = \text{DON, DOP})$$

$$j_{DOC,R} = (1 - \kappa_{E_{CH}})j_{E_i,R} \tag{16}$$

For the rejection flux from $E_{CH}$-reserve, we assume that the flux $j_{DOC,R}$ is entirely fed to the DOC pool.

According to DEB theory product synthesis must be a weighted sum of the basic powers: assimilation, maintenance and growth (Kooijman, 2010; Lorena et al., 2010). To our knowledge no direct evidence exists for DOM excretion related either assimilation or maintenance. However, DOM can be excreted from growing cells in the exponential phase of culture (Myklestad, 2000; Urbani et al., 2005; Lomas et al., 2000; Saad et al., 2016) and the specific DOM release is proportional to the specific growth rate (Myklestad et al., 1989; Underwood et al., 2004). Therefore, we assume that the growth flux contributes to DOM
production at a specific rate

$$j_{\ell,G} = y_{\ell,V} j_{VG} \quad (\ell = \text{DOC, DON, DOP}) \tag{17}$$

where, $y_{\ell,V}$ stoichiometrically couples DOM production to the growth rate according to the relationship: $y_{\ell,V} = y_{E,V}n_{*,E} - n_{*,V}$ ($* = $ C, N, P). Finally, inorganic carbon, ammonium and phosphorus are released as by-products from the maintenance processes according to the stoichiometric relationship

$$j_{i,M} = n_{*,E}j_{E,M} + j_V^M(n_{*,V} - y_{E,V}n_{*,E}) \tag{18}$$

where $i$ = IC, NH, P. Therefore, Eqs. (16) and (17) give the excretion fluxes of DOM $j_{\ell,E} = j_{\ell,R} + j_{\ell,G}$ ($\ell$= DOC, DON, DOP) and Eq. (15) and Eq. (18) give the excretion fluxes of inorganic nutrients $j_{i,E} = j_{i,R} + j_{i,M}$ ($i$=NH,P), while for nitrate $j_{NO,E} = j_{NO,R}$. For the inorganic carbon excretion rate, in addition to $j_{IC,M}$ (Eq. 18) there are two extra sources of carbon production associated with the extra carbohydrates requirements for E and E′ formation depending on the nitrogen source (see
Sect. 2.2.2). Thus, the excretion fluxes of inorganic carbon is $j_{IC,E} = (y_{CH,E} - n_{C,E})j_{E,A} + (y_{CH,E'} - n_{C,E'})j_{E',C} + j_{IC,M}$.

### 2.3   Linking the model to experimental data

The model was first calibrated using published experimental data for *Thalassiosira pseudonana* (Flynn et al., 2008). Data were digitized from Fig. 4 in Flynn et al. (2008) using the WebPlotDigitalizer (Rohatgi, 2017). These data allowed tracking of carbon and nitrogen in the inorganic and organic pools but did not provide any evidence about phosphorus. Thus, we
assumed that growth was not phosphorous limited for this data set. Parameter values, given in Table 2, were taken from the literature and, if not available, they were tuned against the experimental data. Furthermore, molecular elemental ratios of biomass were used in order to constrain the parameter values so that the resulting ratios would fall close to observable ranges. The molecular elemental ratios of biomass are calculated as follows: $n_N = n_{N,E}M_E + M_{E_{NH}} + M_{E_{NO}} + n_{N,V}M_V$ is the total N-mol content, $n_P = n_{P,E}M_E + M_{E_P} + n_{P,V}M_V$ is the total P-mol content and $n_C = n_{C,E}M_E + M_{E_{CH}} + n_{C,V}M_V$
is the total C-mol content of the cells. Thus, C:N $= n_C/n_N$, C:P $= n_C/n_P$ and N:P $= n_N/n_P$. Finally, the method of local sensitivity analysis was used in order investigate the sensitivity of the model to parameter values and the results can be found in the supplementary information (Sect. S3, Table S1). Model simulations were implemented in MATLAB (version R2009b).





## 3 Results

### 3.1 Model validation

The model (Eqs. (1)–(3)) has a very close qualitative and quantitative correspondence with the experimental data (Fig. 2). The total particulate carbon (POC) and nitrogen (PON) in the system comprise the sum of each element in the structural mass, reserves and dead structural mass. Under both nitrate and ammonium growth, POC in the system initially increases exponentially and reaches a stationary phase when nutrients are being consumed and eventually are depleted. Under nitrate-growth (Fig. 2, left), the model predictions for the inorganic (IC) and organic dissolved (DOC) and particulate (POC) carbon match rather well the experimental measurements. The model output for the particulate (PON) and dissolved organic (DON) nitrogen describes adequately the experimental data during the exponential phase, however, in the stationary phase, PON is overestimated while DON is underestimated. Under ammonium-growth (Fig. 2, right), the model predicts well the consumption of inorganic carbon and ammonium and the production of PON and DON. Model predictions for POC and DOC agree with the data up to day 5, but from day 6 onwards the data suggest a continuous increase in POC and a significant decrease in DOC, both of which are not captured by the model.

### 3.2 Model analysis

In order to explore the effects of nutrient limitation on DOM release by phytoplankton, we performed 8-days simulations of Eqs. (1) – (3) under two different nitrogen and phosphorus availability scenarios, using the parameter values given in Table 2. In the N-limited scenario, initial conditions of nitrate and phosphate were set to a ratio of N:P = 5, with initial values $X_{NO}(0) = 60$ μM and $X_P(0) = 12$ μM and in the P-limited scenario were set to a ratio of N:P= 60 ($X_{NO}(0) = 120$ μM and $X_P(0) = 2$ μM). In both scenarios, the initial ammonium concentration in the medium was set equal to zero and the inorganic carbon concentration was kept constant at 2000 μM to avoid carbon limitation. The initial values for the rest of the variables were the same for both scenarios ($X_V(0) = 35$ μM, $X_E(0) = 10$ μM, $X_{E_{CH}}(0) = 5$ μM, $X_{E_{NH}}(0) = X_{E_{NO}}(0) = 1.73$ μM, $X_{E_P}(0) = 0.4083$ μM, $X_{DOC}(0) = X_{DON}(0) = X_{DOP}(0) = 0$ μM).

#### 3.2.1 Biomass, Primary production and DOC release

The temporal patterns of inorganic nitrogen, which is the sum of ammonium and nitrate concentrations in the environment, inorganic phosphorus and biomass are presented in Fig. 3. Biomass concentration is measured in terms of carbon in structural mass and in E- and $E_{CH}$-reserve. Structural mass-specific metabolic rates are presented in Fig. 4. The net growth rate, $r$, (see Sect. 2.2.3) is responsible for the production of new structural mass, after maintenance and growth costs having been paid (Fig. 4a). The carbohydrates assimilation flux $j_{CH,A}$, given in Eq. (9), relates to the photosynthetically produced organic carbon and corresponds to primary production (PP) (Fig. 4b). Specific DOC release rate corresponds to the sum of fluxes that contribute to the release of dissolved organic carbon from the cell (Fig. 4c). Percentage Extracellular Release (PER) is calculated as the percentage of DOC release over PP (Fig. 4d). The temporal patterns of structural mass-specific densities of





nitrogen ($m_{E_{NO}} + m_{E_{NH}}$) phosphorus ($m_{E_P}$) carbohydrates ($m_{E_{CH}}$) and generalized ($m_E$) reserves are presented in Fig. 5 and are used to explain the observed dynamics.

During the 8-days simulations, three phases of nutrient availability are distinguished: the nutrient-replete, the intermediate and the nutrient-limited phase, that correspond, roughly, to three phases of phytoplankton growth; the initial, the exponential

and the stationary, respectively. The end of nutrient-replete phase is marked when >99 % of the initial value of the limiting nutrient was consumed; around day 3.5 in the N-limited scenario (Fig. 3a) and day 4 in the P-limited scenario (Fig. 3b). The end of the intermediate phase is marked by the maximum value of biomass; around day 6 in the N-limited scenario (Fig. 3c) and day 7 in the P-limited scenario (Fig. 3d). In both scenarios, biomass increases during the nutrient-replete and intermediate phases. In addition, biomass, increases approximately 1.5-fold more in the P-limited compared to the N-limited scenario (Fig. 3c,d).

The higher biomass observed in the P-limited scenario, is because P-limited cells maintain higher specific growth rate (Fig. 4a) and higher specific PP rate (Fig. 4b) than N-limited cells during the intermediate and the early nutrient-limited phase. However, in the late nutrient-limited phase of the P-limited scenario, the low densities of nitrogen and phosphorus reserves (Fig. 5c,d, red asterisks), that coincide with the lowest specific PP and DOC release rates (Fig. 4b,c, red asterisks) observed overall, suggest that cells are co-limited by both nutrients.

Generally, in both scenarios, specific DOC release rate presents a maximum value in the intermediate phase and a minimum value in the nutrient-limited phase (Fig. 4c). DOC release rate has the same pattern as the carbohydrates reserve density ($m_{E_{CH}}$) dynamics (Fig. 5b). Overall, the highest specific DOC release rate is observed in the late intermediate phase in the P-limited scenario (Fig. 4c) and coincides with the overall highest carbohydrates reserve density (Fig. 5b). Model predictions for both scenarios suggest that PER remains constant at around 8 % during the nutrient-replete period (Fig. 4d, solid lines). During

the intermediate phase, PER rapidly increases (Fig. 4d, dash-dot lines) to reach its maximum value within the nutrient-limited phase (53 % in the N-limited scenario, 64.5 % P-limited scenario) (Fig. 4d, asterisks). Comparing the two scenarios, in the intermediate phase and early nutrient-limited phase, PER is higher in the N-limited scenario, while, overall, the highest PER is observed in the late nutrient-limited phase in the P-limited scenario (Fig. 4d).

DOC release rate is analyzed into the contributing fluxes that are presented in Fig. 6. DOC release has contributions from

three fluxes: $j_{DOC,R}$, which stands for the active exudation of DOC due to nutrient limitation, given in Eq. (16), $j_{DOC,G}$, which is proportional to growth rate, given in Eq. (17), and the flux $j_{DOC,D} = h(X_E + X_{E_{CH}})$ that relates to the release of DOC due to cell death and subsequent lysis. In both scenarios, $j_{DOC,G}$, has the highest contribution to DOC release during the nutrient-replete phase (Fig. 6, dashed line). In the intermediate phase the relative contribution of $j_{DOC,G}$ to the DOC release flux decreases, while the relative contribution of $j_{DOC,R}$ flux increases to become the most significant flux of DOC release

during the nutrient-limited phase (Fig. 6, solid line). The relative contribution of $j_{DOC,D}$ flux to total DOC release rate is always less than 10% (Fig. 6, dash-dot line). In the N-limited scenario the depletion of extracellular inorganic nitrogen, which marks the onset of the intermediate phase, coincides with the increasing relative importance of the $j_{DOC,R}$ flux (Fig. 6, left). In the P-limited scenario, the relative importance of the $j_{DOC,R}$ for the DOC release increases earlier, in the late nutrient-replete phase (Fig. 6, right). This indicates that the cells become P-limited from the early stage in the P-limited scenario, which is





also evident from the significant decrease of $m_E$ and $m_{E_P}$ in the second half of the nutrient-replete phase (Fig. 5a,d) and the corresponding increase of the non-limiting $m_{E_{NO}} + m_{E_{NH}}$ and $m_{E_{CH}}$ (Fig. 5b,c).

The relationship between total DOC release and total primary production rate, is presented in Fig. 7. Total DOC release and PP rates in the model are calculated by multiplying the specific DOC release and the specific PP rate with $X_V$. In both

scenarios, during the nutrient-replete phase there is a positive linear relationship between DOC release and PP rates (Fig. 7, solid lines). During the intermediate phase, the linearity between PP and DOC release is lost as PP rate slows down while DOC release rate increases (Fig. 7, dash-dot lines). The decrease in PP rate is due to the effects of nutrient limitation on the photosynthetic rate (see Sect. 2.2.1). In the N-limited scenario, the linearity is lost as soon as cells enter the intermediate phase (Fig. 7, dash-dot blue line) while in the P-limited scenario, the linearity is lost later in the intermediate phase (Fig. 7, dash-

dot red line). This suggests that the effects of nutrient limitation on PP rate are stronger under nitrogen-limiting conditions than under phosphate-limiting conditions. In both scenarios, the increase in DOC release rate is due to the increase of the $j_{DOC,R}$ flux (Fig. 6, solid line) and it is indicative of stoichiometrically unbalanced growth. In the nutrient-limited phase (both scenarios), total PP rate further decreases (Fig. 7, asterisks). As a result, $m_{E_{CH}}$ decreases (Fig. 5b, asteriks) and this leads to the decrease of the $j_{DOC,R}$ flux and, therefore, to the decrease of the DOC release rate (Fig. 7, asterisks). Comparing total

DOC release with total net primary production, which in the model is calculated by multiplying the specific net growth rate, $r$, with $X_V$, the same patterns described above for total DOC release and PP rates are observed in both scenarios (figure not shown).

### 3.2.2    Elemental ratios of phytoplankton and DOM released

Elemental ratios of C, N and P of phytoplankton biomass and of produced DOM are shown in Fig. 8. During the nutrient-replete

phase, elemental ratios of biomass are relatively constant (Fig. 8a–c), with the exception of the late nutrient-replete phase in the P-limited scenario, where C:P and N:P ratios increase. At the onset of the intermediate phase and the depletion of nutrients, C:N ratio rapidly deviates to reach its maximum at the end of the intermediate phase, while C:P ratio has a rapid increase in the P-limited scenario, but no considerable deviation in the N-limited scenario. C:N and C:P ratios reach their highest value in the N-limited and P-limited scenarios, respectively. In addition, the biomass N:P ratio during the nutrient-limited phase has an

overall maximum value in the P-limited scenario and an overall minimum value in the N-limited scenario. The variations of elemental ratios are due to the variations of reserve densities (Fig. 5)

The elemental ratios of DOM released by phytoplankton present the same trends as the cellular ratios in both scenarios. Thus, constant values during the nutrient-replete phase (Fig. 8d–f) are observed, as DOC, DON and DOP released by phytoplankton are related to growth (see Sect. 2.2.4, Eq. (17)), therefore, displaying a constant stoichiometry. In the beginning of the

intermediate phase, the composition of released DOM is no longer constant. The DOC release rate is higher than the DON and DOP release rates as their constituting elements, N and P, are limiting and, therefore, are not released in high rates. In the N-limited scenario, the largest deviations from the elemental ratios of the nutrient-replete phase are observed for the DOC:DON ratio, while, in the P-limited scenario the largest deviations are observed for the DOC:DOP ratio. Finally, DON:DOP has an overall maximum value in the P-limited scenario and an overall minimum value in the N-limited scenario.





### 3.2.3 Molecular composition and size of DOM released

In order to further investigate the quality and molecular size of DOC produced by phytoplankton in our model, we assigned the released DOC into two separate groups, namely $DOC_L$ and $DOC_H$. We defined as $DOC_L$ the fraction of DOC released in association with the processes of growth ($j_{DOC,G}$, Fig. 6, dashed line) and death ($j_{DOC,D}$, Fig. 6, dash-dot line), which

reflect the passive diffusion of small (<10 kDa) mono and polysacharides, rich in glucose and/or the DOC fraction of the dissolved aminoacids or other phosphorus-containing organic molecules, (Bjørnsen, 1988; Urbani et al., 2005; Flynn et al., 2008; Borchard and Engel, 2015). Moreover, $DOC_H$, is associated with $j_{DOC,R}$ flux (Fig. 6, solid line) and reflects the active exudation of DOC due to nutrient limitation, mainly composed of high molecular weight carbohydrates (>10kDa) such as heteropolysacharides or organic carbon associated with the organic forms of nitrogen and phosphate exuded by the cell due to

unbalanced growth (Fogg, 1983; Urbani et al., 2005; Flynn et al., 2008; Borchard and Engel, 2015). According to model results, during the nutrient-replete phase, in both scenarios, the main fraction of total DOC (TDOC) is almost entirely comprised by low molecular weight $DOC_L$ (mean throughout the nutrient-replete phase: 95 %, Fig. 9). As nutrient limitation progresses, the fraction of high molecular weight DOC, $DOC_H$, progressively increases, as a result of the increased relative importance of the $j_{DOC,R}$ flux (Fig. 6). At the end of the nutrient-limited phase, TDOC is composed by 60.5% and 39.5% of $DOC_H$ and $DOC_L$,

respectively, in the N-limited scenario and 52% and 48% of $DOC_H$ and $DOC_L$, respectively, in the P-limited scenario.

## 4 Discussion

In this study, we have developed a dynamic model of phytoplankton growth in order to elucidate the physiological mechanisms that govern the exudation of dissolved organic matter during the different growth phases of phytoplankton. The model explicitly includes inorganic nutrients and carbohydrates reserves, therefore allowing to follow the variable biomass stoichiometry. The

assimilation of carbon and inorganic nutrients was modelled based on the kinetics of the synthesizing unit. This approach enabled us to take into account multiple nutrient limitation of growth, by nitrogen and phosphorus, and also to account for the interactions between nitrate and ammonium assimilation by phytoplankton. Furthermore, being based on the general framework for metabolic organization, provided by DEB theory, the model, although calibrated against the diatom *T. pseudonana*, can potentially be applied to describe the growth dynamics of any phytoplankton species.

### 4.1 Model performance

Overall, the model was in good agreement with the data retrieved from Flynn et al. (2008). However, under nitrate-growth in the stationary phase PON is overestimated while DON is underestimated, compared to the experimental data. Given that there was not any unexplained change in the carbon compartment, we hypothesize that the decrease of PON and the parallel increase in DON in the experimental data could be related to the breakdown of nitrogenous organic compounds to simpler molecules

that were measured in the DON compartment; a process that is not included in the model. Under ammonium-growth, during the stationary phase, the increase in POC and the significant decrease in DOC observed in the data were not reproduced by the



model. A possible explanation, would be the coagulation of dissolved polysaccharides (DOC) to form transparent exopolymer particles (TEP) that would be measured as POC, as has been observed previously in mesocosms (Schartau et al., 2007) and cultures of *T. pseudonana* (Urbani et al., 2005).

Model predictions regarding elemental biomass composition were in agreement with experimental evidence (Perry, 1976; Goldman et al., 1979; Flynn et al., 2008) (Fig. 8a–c). However, in the P-limited scenario, model predictions overestimated the C:P ratio, reported in Perry (1976), during the late intermediate and the nutrient-limited phase. This could be attributed to the fact that the maximum C:P ratio reported in Perry (1976) was measured in a P-limited chemostat under steady state conditions while, our model is dynamic in time. Nevertheless, when the model was run until steady–state was reached (not shown), C:P ratio eventually matched the experimentally measured C:P ratio.

Local sensitivity analysis revealed that the model outputs of interest, DOC release rate and biomass, were most sensitive to parameters related to photosynthesis light and dark reactions and to stoichiometric coefficients coupled to carbohydrates reserves, during all three nutrient availability phases. Fewer parameters cause changes to model outputs at particular nutrient availability phases. The results of the sensitivity analysis can be found in the supplementary information (Sect. S3, Table S1).

### 4.2  Primary production and DOC release under N- and P-limitation

Generally, higher specific growth and PP rates were observed under P-limiting, than N-limiting conditions, while the non-limiting $E_{CH}$-reserve reached highest densities in the intermediate phase of the P–limiting scenario. These findings suggest that phosphorus deficiency has less severe effect on phytoplankton growth and PP rate than nitrogen deficiency. In accordance to our findings, lower photosynthetic performance has been observed in microalgae under N-limited than P-limited conditions (Kamalanathan et al., 2016; Alcoverro et al., 2000), while in the diatom *Cylindrotheca closterium* carbohydrates content was higher under P-limited growth than under N-limited growth (Alcoverro et al., 2000).

Irrespective of the limiting nutrient, our results showed that DOC release rate followed closely the carbohydrates reserve density dynamics. DOC was produced at high and steady structural mass-specific rates during the nutrient-replete period of growth. Specific DOC release rate had a maximum value during the intermediate phase and a minimum value in the nutrient-limited phase as a result of the reduced photosynthetic rate (PP) during this phase, that fuels the $E_{CH}$-reserve. Conversely, PER, calculated as DOC release over PP rate, had lower values (8%) in the nutrient-replete phase and higher values, 53% and 64.5% in the nutrient-limited phase of the N-limited and P-limited scenario, respectively, as a result of the progressive reduction of PP with the development of nutrient limitation.

The pattern of PP and DOC release rates and the range of PER values predicted by our model is in accordance with many experimental and field studies. For example, in nutrient-limited diatom cultures, maximum extracellular production of carbo-hydrates has been observed at the transition phase between the exponential and the stationary phase (Alcoverro et al., 2000; Urbani et al., 2005), while higher specific (per cell) carbohydrates release rate has been observed in the exponential phase than in the stationary phase (Myklestad, 1995; Myklestad et al., 1989; Granum et al., 2002; Flynn et al., 2008). Furthermore, in a study with the diatom *Chaeloceros affinis* it was found that the specific photosynthetic rate was reduced by 50 % in the transition phase to reach 10 % of the exponential phase value during the stationary phase (Myklestad et al., 1989). More-





over, PER has been found to vary between 2–10 % in the exponential phase and to reach up to 60 % in the stationary phase, nutrient-limited phase, depending on the species and the condition of the culture (Myklestad, 2000). Field studies have also reported average PER values <10 % in productive areas under nutrient-replete conditions and average values up to 46 % under oligotrophic conditions (Teira et al., 2001a, b; Lagaria et al., 2013).

The relationship between primary production and DOC release is of great interest in the field studies, but to date it hasn't been fully understood since, based on empirical and experimental observations, both linear and non-linear relationships have been observed. For example, Teira et al. (2001a) showed that there was a linear 1:1 relationship between log transformed primary production and DOC release rates for three upwelling regions (Benguela (SW Africa), Mauritania (NW Africa) and NW Spain) while for data from the oligotrophic North Atlantic subtropical gyre no significant relationship was found. López-

Sandoval et al. (2011) have reported a significant linear relationship in the oligotrophic Mediterranean basin while Lagaria et al. (2011) have reported a moderate relationship of slope <1 during microcosm experiments with nutrient additions in surface waters of the oligotrophic Mediterranean Sea. Our model study revealed that these two rates, DOC release and PP rates, present a linear relationship only in the nutrient-replete phase of growth while this relationship is lost during the intermediate and the nutrient-limited phase, for both scenarios, due to the different mechanisms that drive the dynamics of the two rates. Under this

point of view, the variability found in field data may be better understood. For example, in the results of Lagaria et al. (2011), a strong linear relationship may be observed when the data corresponding to the double nitrogen and phosphorus additions are considered, while no apparent relationship is observed, when taking into account the single nutrient additions, since the system was most probably co-limited by nitrogen and phosphorus.

### 4.3   Mechanisms of DOM production

In this study, using the DEB model for phytoplankton (Kooijman, 2010), we were able to discriminate between the two major conceptual mechanisms of DOM release, i.e., passive diffusion and active exudation, in contrast to existing phytoplankton models that involve DOM exudation (Van Den Meersche et al., 2004; Schartau et al., 2007; Flynn et al., 2008; Kreus et al., 2014) but do not discriminate between the mechanisms of DOM release. We assumed that the DOC associated with growth and death processes will be small in size and will be released via the passive diffusion mechanism while high molecular weight

(HMW) DOC will be associated with the mechanism of rejection of unprocessed substrates by the SU and, thus, with the active exudation mechanism. Furthermore, we assumed that the two types of DOC produced by phytoplankton (i.e., $DOC_L$, $DOC_H$), will have different molecular composition. Since $DOC_L$ is excreted via passive diffusion and cell lysis, it is expected to have a similar composition as the cellular material (Borchard and Engel, 2015), while $DOC_H$ excreted via active exudation is expected to have a more distinct composition from the cellular material (Biersmith and Benner, 1998; Borchard and Engel,

2015). As such, the model suggests that the relative importance of the two mechanisms and, thus, the relative presence of high and low molecular weight carbohydrates with different molecular composition signatures, is dependent on the nutrient status of the cells. DON and DOP are produced by the cells in an analogous way.

In accordance with our suggestion, it was found that in steady-state, P-limited cultures of *Emiliania huxleyi* glucose was the dominant monomer in both the small size fraction (1–10 kDa) of dissolved polysacchrides and in the particulate fraction





(cell content) and less significant in the larger size fractions (>10 kDa) (Borchard and Engel, 2015). In contrast, arabinose and galacturonic acid were the most abundant monomers in the larger size fractions and contributed less to the small and particulate size fractions (Borchard and Engel, 2015). In cultures of marine diatoms, glucose has been identified as the most abundant monomer during the exponential, nutrient-replete phase of growth in the extracellular carbohydrates. A pronounced

decrease of glucose and an increase of heteropolysacharides, containing various monomers, has been observed in the stationary, nutrient-limited phase (Underwood et al., 2004; Urbani et al., 2005). The molecular composition of DOM can have implications for its subsequent utilization by bacteria. Many studies have shown that exudates, rich in glucose, are taken up rapidly by bacteria while heteropolysaccharides, consisting of a variety of monomers, have been found to escape bacterial degradation (Obernosterer and Herndl, 1995; Hama and Yanagi, 2001; Puddu et al., 2003).

In addition, the elemental composition of produced DOM can be highly variable depending on the physiological status of phytoplankton and the limiting nutrient. Depending on the limiting nutrient, N or P, the model's output suggests increased DOC:DOP or DOC:DON ratios in the intermediate and nutrient-limited phase, compared to the nutrient-replete phase. These findings qualitatively agree with experimental results from phytoplankton cultures (Biersmith and Benner, 1998; Saad et al., 2016), mesocosm studies (Engel et al., 2002; Conan et al., 2007) and oceanic regions (Saad et al., 2016 and references therein),

that report increased DOC:DOP and DOC:DON ratios upon phosphate or nitrogen limitation, respectively. Furthermore, Hopkinson and Vallino (2005) estimated the mean labile DOM C:N ratio to be 10.7, C:P to be 199 and N:P 20. These values are comparable to the model predictions during the nutrient-replete phase under both nutrient availability scenarios. On the other hand, refractory DOM is carbon-rich and depleted in nitrogen and phosphorus (Hopkinson and Vallino, 2005) as was predicted by the model during the nutrient-limited phase.

By taking together the above mentioned model results and findings reported in the literature, we suggest that during the phase of unlimited growth in nutrient-replete cultures or oceanic regions, labile DOM is produced with low C:N and C:P ratio while the carbohydrates fraction of DOC produced is rich in glucose. Under these conditions, the physiological mechanism responsible for DOM production is mainly related to growth rate of phytoplankton and the passive diffusion hypothesis (Bjørnsen, 1988; Mueller et al., 2016). This organic matter is efficiently utilized by bacteria (Puddu et al., 2003; Hama and Yanagi, 2001).

As a result, a tight coupling is expected between phytoplankton production and bacterial production. This expectation is actually confirmed from field studies that report a direct coupling of phytoplankton–bacteria dynamics in productive, upwelling ecosystems (Teira et al., 2015; Wear et al., 2015) or in microcosms after nutrient additions (Lagaria et al., 2011; Fouilland et al., 2014). On the contrary, under nutrient limiting conditions, DOM produced will be rather refractory, depleted in N and/or P and rich in heteropolysacharides that escape bacterial degradation and can accumulate in the water column (Obernosterer

and Herndl, 1995; Thingstad et al., 1997; Hama and Yanagi, 2001; Saad et al., 2016). We propose that this DOM is associated with the overflow model and the rejection flux of the non-limiting compounds, i.e., organic carbon (Fogg, 1983; Mueller et al., 2016). Consequently, the lack of correlation between phytoplankton PP or DOC release and bacterial production, often documented in nutrient poor conditions (Lagaria et al., 2011; Fouilland et al., 2014; Teira et al., 2015), may be related to the prevalence of the rejection flux related to the overflow mechanism.



## 5  Conclusions

In summary, we developed a dynamic model explicitly describing the DOM release by phytoplankton under nitrogen and phosphorus limitation. The model allows for the distinction and assessment of the two major mechanisms of DOM release: the passive diffusion which is related to growth and lysis of the cells and the active exudation which is related to rejection of

5   unprocessed substrates by the SU. Passive diffusion of lower molecular weight DOM, rich in glucose, contributes more under nutrient-replete conditions and the DOM produced has a stoichiometry representative of labile DOM. Under nutrient-limiting conditions, active exudation is the mechanism mainly responsible for the production of DOM which is mostly composed of non-labile heteropolysacharides of high molecular weight and with elemental ratios representative of refractory DOM. Our model, by allowing to discriminate between the two mechanisms of DOM production, suggests that the decoupling of primary

10   production and DOC release and of primary production and bacterial production, often observed in oligotrophic conditions is related to the prevalence of the active exudation mechanism. Our model can be coupled with biogeochemical models to resolve the bioavailability of DOM produced and, therefore, allow for better description and forecast of the carbon cycling in the euphotic layer and export in the deep sea.





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





**Table 1.** Table of frequently used symbols. Dimensions: $t$ means time and $l$ length.

| Symbol | Interpretation (Dimensions) |
|---|---|
| $t$ | Time ($t$) |
| $M_V, M_{E_i}$ | Mass of structure, reserves (mol ) |
| $m_{E_i}$ | Density of reserve $E_i = M_{E_i}/M_V$ (mol/mol) |
| $X_i$ | Concentration of compound $i$ (mol $i\ l^{-3}$) |
| $I$ | Incident light intensity (mol photons $l^{-2}\ t^{-1}$) |
| $j_{i,k}$ | Specific flux of compound $i$ associated with process $k$ (mol $i$ (mol $V$)$^{-1}\ t^{-1}$) |
| $a_L$ | Specific photons' arrival cross section ($l^2$ (mol $V$)$^{-1}$) |
| $a_i$ | Specific arrival rate of compound $i$ ($l^3$ (mol $V$)$^{-1}\ t^{-1}$) |
| $\rho_i$ | Binding probability of compound $i$(–) |
| $K_i$ | Half-saturation constants for compound $i$ (mol $i\ l^{-3}$) |
| $k_i$ | Handling/turnover rate of compound $i$ ($t^{-1}$) |
| $\kappa_{E_i}$ | Fraction of rejected flux of $E_i$ returning to reserves (–) |
| $y_{i,j}$ | Stoichiometric coefficients (mol $i$ (mol $j$)$^{-1}$) |
| $n_{*_1,*_2}$ | Number of atoms of element $*_1$ present in compound $*_2$ (#/#) |
| $j_{E_M}$ | Maintenance rate ($t^{-1}$) |
| $h$ | Death rate ($t^{-1}$) |
| $\beta_{DON}$ | fraction of dead $E_{NH}$ directed to DON |





**Table 2.** Table of parameter values.

| Arrival parameters (mol (mol C)$^{-1}$d$^{-1}$) or (m$^2$(µmol C)$^{-1}$ for $a_L$, L (µmol C)$^{-1}$d$^{-1}$ for $a_{NO}$) | | | | | | | |
|---|---|---|---|---|---|---|---|
| $a_L$ | $6*10^{-6}$ | $j_{IC_{max}}$ | 5.1 | $j_{P_{max}}$ | 0.01 | $j_{NH_{max}}$ | 0.15 |
| $a_{NO}$ | 0.6 | | | | | | |
| **Handling rates ($k$, d$^{-1}$) and Half saturation constants ($K$, µmol L$^{-1}$)** | | | | | | | |
| $k_L$ | 100 | $k_{CH}$ | 20 | $k_1$ | 0.15 | $k_2$ | 1 |
| $k_{NH}, k_{NO}$ | 20 | $K_{IC}$ | 663 | $K_{NH}$ | 0.5 [1] | $K_P$ | 0.5 [2] |
| **Binding probabilities (dimensionless)** | | | | | | | |
| $\rho_{L_{max}}$ | 0.9 | $\rho_{L_{min}}$ | 0.03 | $\rho_*$ | 0.9 | $\rho_1, \rho_2$ | 0.7 |
| **Turnover rates (d$^{-1}$)** | | | | | | | |
| $k_E$ | 3.6 | $k_{E_{CH}}$ | 2.6 [3] | $k_{E_{NO}}$ | 0.9 | $k_{E_{NH}}$ | 0.9 |
| $k_{E_P}$ | 0.9 | $j_{E,M}$ | 0.1 | $h$ | 0.01 | | |
| **Stoichiometric coefficients (mol mol$^{-1}$)** | | | | | | | |
| $y_{NA,L}$ | 0.2 [4] | $y_{NA,E_{CH}}$ | 2 | $y_{IC,E_{CH}}$ | 1 | $y_{CH,E}^{NO}$ | 1.8 |
| $y_{CH,E}^{NH}$ | 1.55 | $y_{N,E}$ | 0.08 | $y_{P,E}$ | 0.00154 | $y_{E,V}$ | 1.2 |
| $n_{P,E}$ | 0.00154 | $n_{P,V}$ | 0.00124 [5] | $n_{N,E}$ | 0.08 | $n_{N,V}$ | 0.079 [6] |
| **Other parameters (dimensionless except I)** | | | | | | | |
| $\kappa_{E_*}$ | 0.95 | $\beta_{DON}$ | 0.5 | $\kappa_{X_i}$ | 0.5 | $b_N$ | 1.6 |
| $v_{P_{max}}$ | 1 | $v_{P_{min}}$ | 0.1 | $b_P$ | 0.45 | I | 100 µmol hv m$^{-2}$ s$^{-1}$ |

Parameters derived from: [1] Anderson and Williams (1998), [2] Perry (1976), [3] Lorena et al. (2010), [4] Lika and Papadakis (2009), [5] Pahlow and Oschlies (2009), [6] estimated from Gallager and Mann (1981)




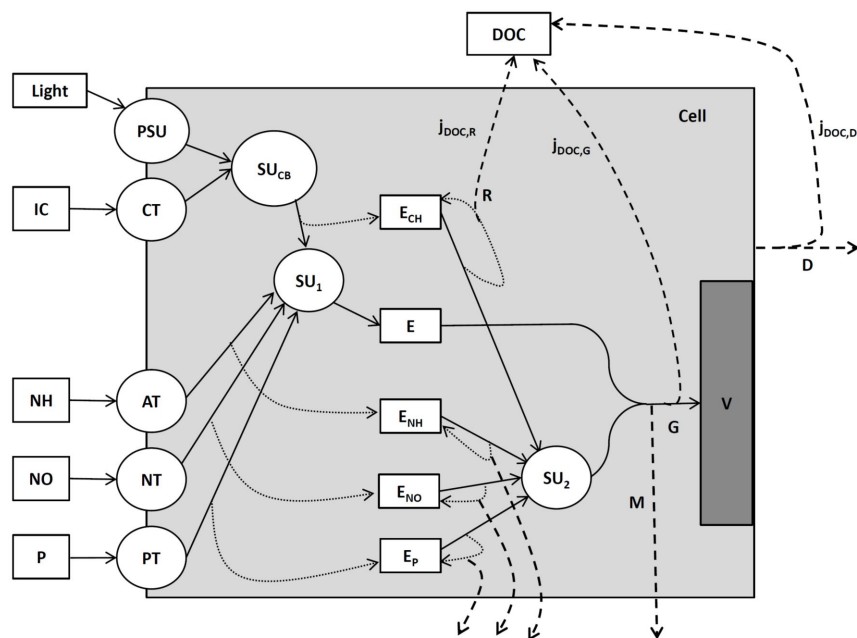

**Figure 1.** Schematic presentation of the model. Boxes represent concentrations, circles synthesizing units/transporters and arrows mass fluxes. CT, AT, NT, PT : carbon-, ammonium-, nitrate-, phosphorus- transporter. $SU_{1,2}$ synthesizing units/generalized enzymes, PSU: photosynthetic unit, $SU_{CB}$ : synthesizing unit for Calvin–Benson cycle, $E_i$ : reserves, V: structure. G, M, R and D denote, respectively, growth, maintenance, rejection and death. $j_{DOC,k}$ denotes the DOC flux associated with process $k$ ($k$ = G, R, D). Only the contributing fluxes to DOC are illustrated. DON and DOP pools and the contributing fluxes similar to that of DOC are also considered (see text for explanation). Solid arrows denote incoming fluxes, dotted arrows denote rejected fluxes fed to the reserves, dashed arrows denote fluxes fed to the environment.





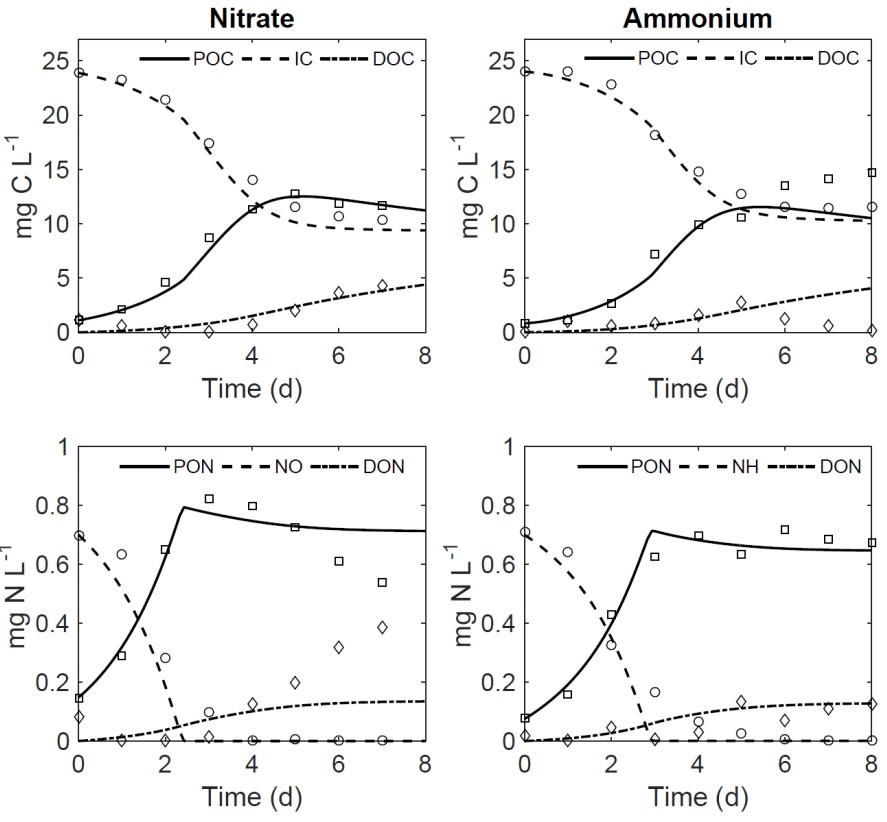

**Figure 2.** Model fits to data from Flynn et al. (2008) for *Thalassiosira pseudonana*. Data as symbols, model output as lines; IC: dissolved inorganic carbon, POC: particulate organic carbon, DOC: dissolved organic carbon, NO: nitrtae, NH: ammonium, PON: particulate organic nitrogen, DON: dissolved organic nitrogen.





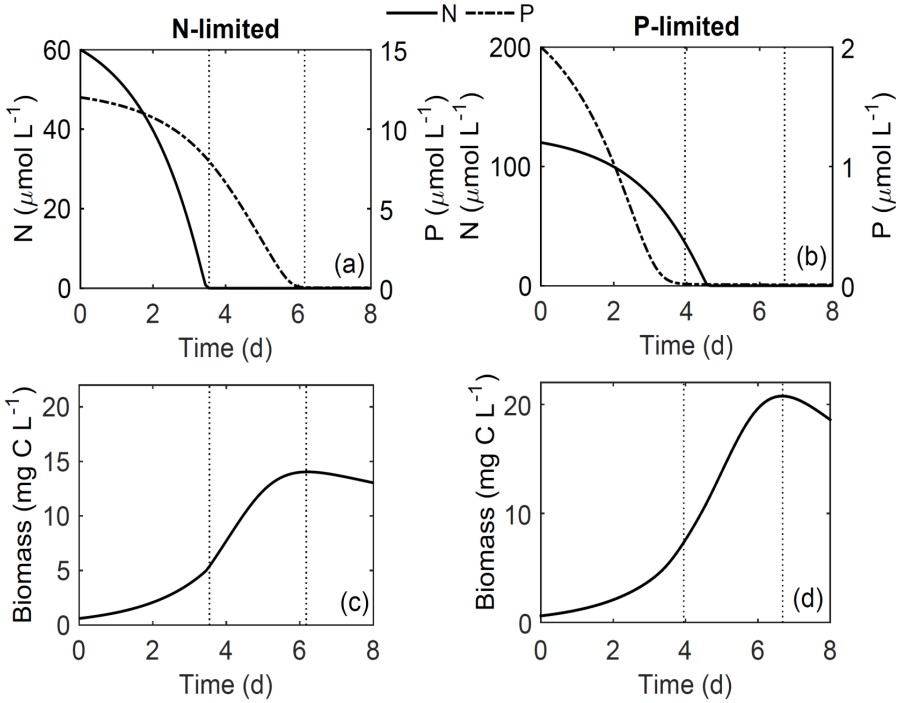

**Figure 3.** Time-course concentration profiles for the inorganic nutrients (top; inorganic nitrogen, N, (solid line), inorganic phosphorus, P, (dash-dot line) and biomass (bottom) for the N-limited scenario (left) and P-limited scenario (right). Vertical dotted lines denote the three phases of nutrient availability (nutrient-replete, intermediate, nutrient-limited)

.



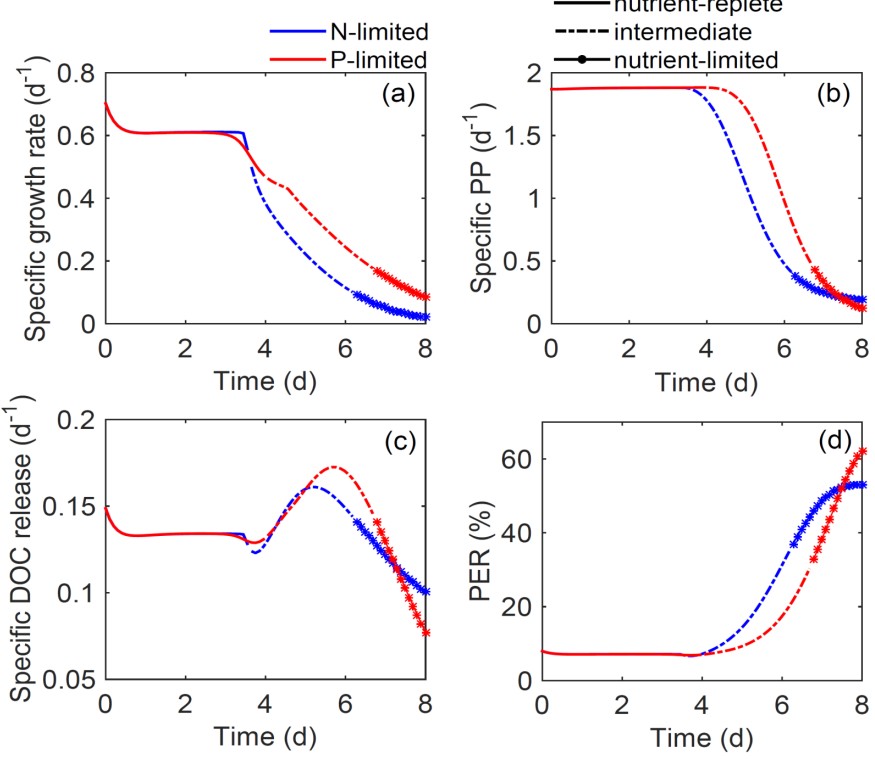

**Figure 4.** Structural mass-specific rates. (a) specific growth rate, (b) specific primary production (PP) rate, (c) specific DOC release rate and (d) Percentage Extracellular Release (PER) for the N-limited (blue line) and the P-limited scenario (red line). Solid lines denote the nutrient-replete phase, dash-dot lines denote the intermediate phase and asterisks denote the nutrient-limited phase.





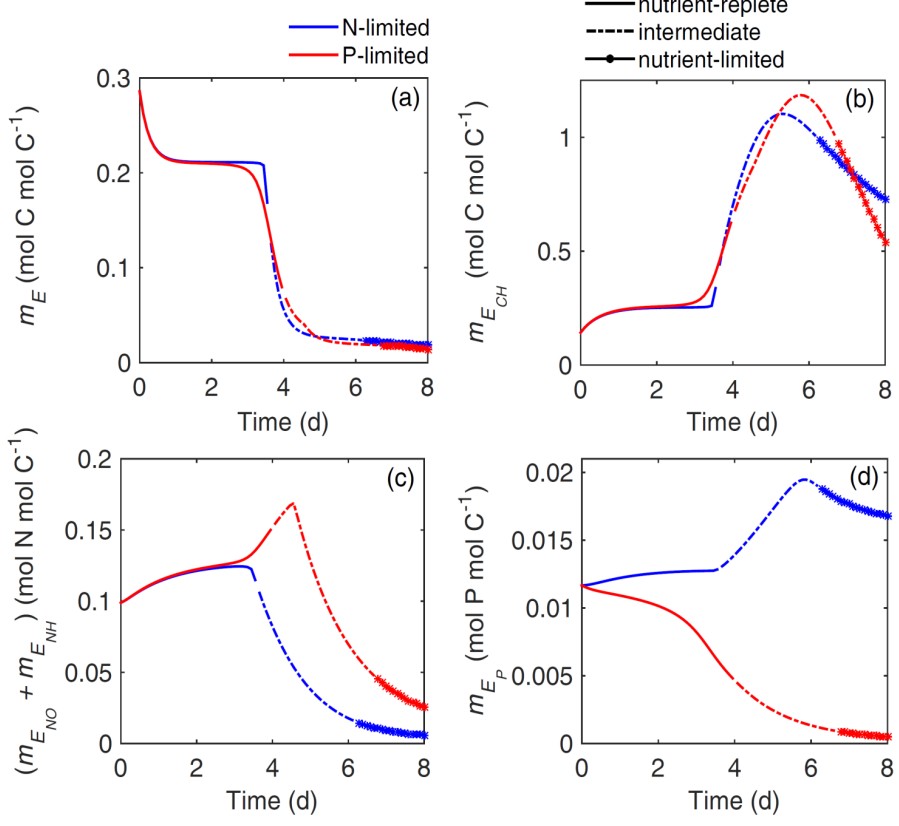

**Figure 5.** Time-course profiles of the structural mass-specific reserve densities, for the generalized reserve (a), the carbohydrates reserve (b), the nitrogen reserves ($E_{NO} + E_{NH}$) (c) and the phosphorus reserve (d) for the N-limited (blue line) and the P-limited (red line) scenario. Solid lines denote the nutrient-replete phase, dash-dot lines denote the intermediate phase and asterisks denote the nutrient-limited phase.





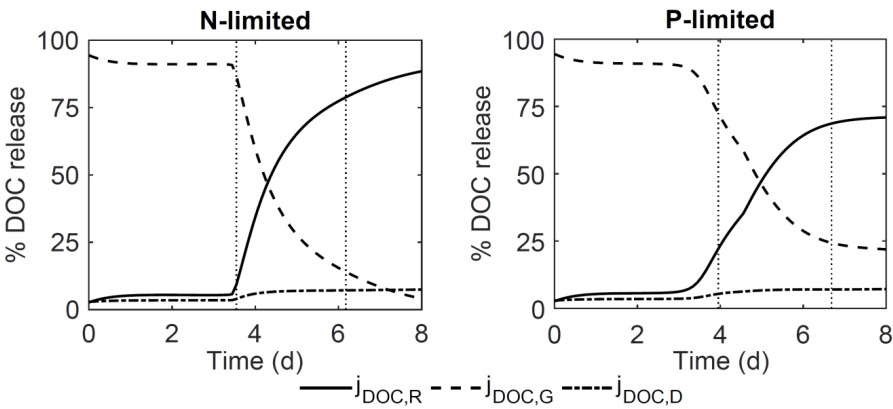

**Figure 6.** Analysis of the DOC release rate into the contributing fluxes for the N-limited (left) and the P-limited (right) scenario ($j_{DOC,G}$: dashed line, $j_{DOC,R}$: solid line, $j_{DOC,D}$: dash-dot line). Vertical dotted lines denote the three phases of nutrient availability (nutrient-replete, intermediate, nutrient-limited)

.



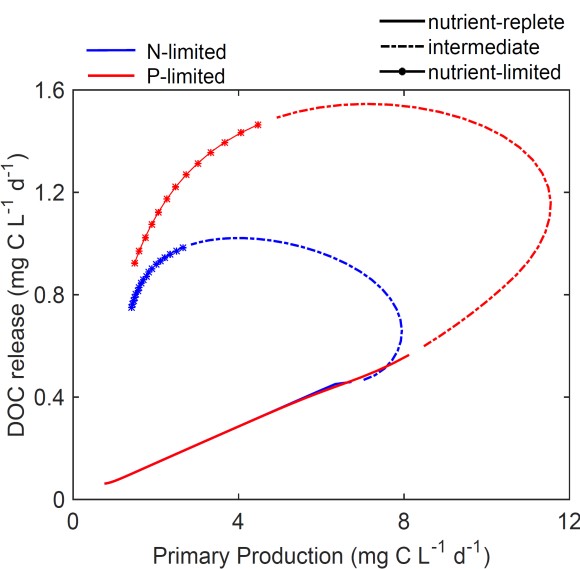

**Figure 7.** Relationship between total DOC release rate and total PP rate for the N-limited (blue line) and the P-limited scenario (red line). Solid lines denote the nutrient-replete phase, dash-dot lines denote the intermediate phase and asterisks denote the nutrient-limited phase.



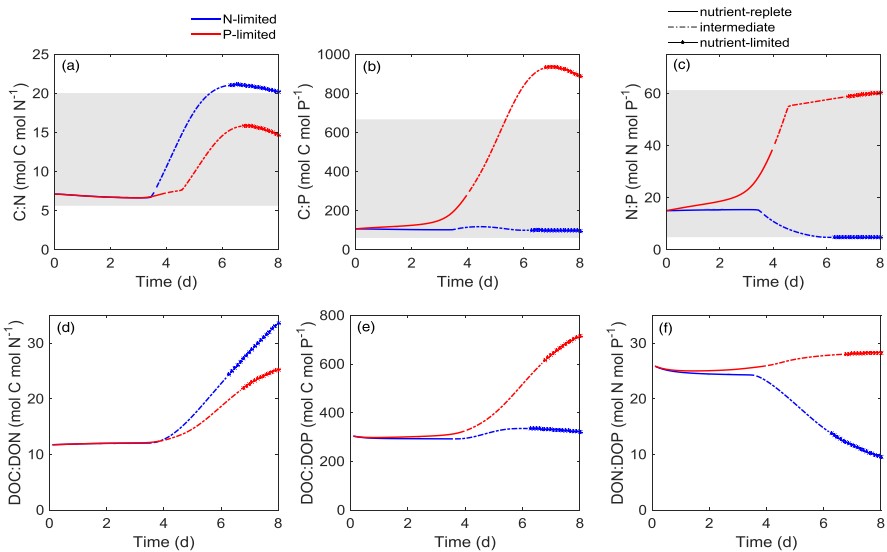

**Figure 8.** Time-course evolution of the biomass C:N (a), C:P (b), N:P (c) ratios and the released dissolved organic matter DOC:DON (d), DOC:DOP (e), DON:DOP (f) ratios for the N-limited (blue line) and the P-limited scenario (red line). Solid lines denote the nutrient-replete phase, dash-dot lines denote the intermediate phase and asterisks denote the nutrient-limited phase. Shaded areas in (a–c) represent the range of elemental ratios observed in the literature for the species *Thalassiosira pseudonana* (Perry, 1976; Goldman et al., 1979; Flynn et al., 2008)

.





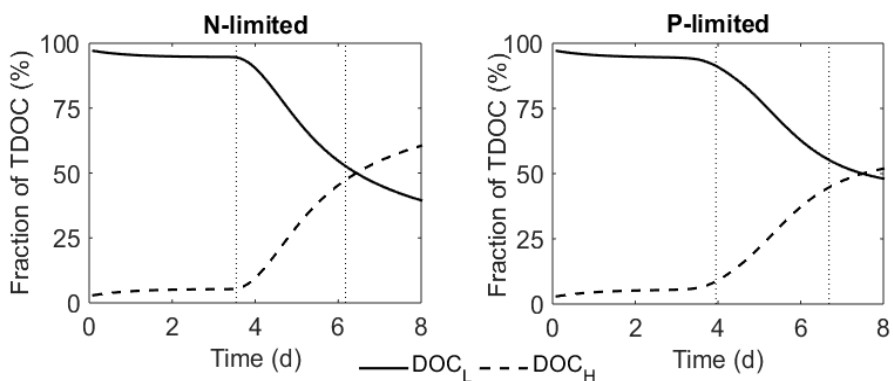

**Figure 9.** Distribution of the two size fractions of DOC produced by phytoplankton in the N-limited (left) and the P-limited (right) scenario. $DOC_L$ denotes the low molecular weight DOC (dashed line), $DOC_H$ denotes the high molecular weight DOC (solid line) (see text for explanation). Vertical dotted lines denote the three phases of nutrient availability (nutrient-replete, intermediate, nutrient-limited)

.