# Peer review of "Dissolved organic matter release by phytoplankton in the context of the Dynamic Energy Budget theory"

_Biogeosciences, 2017_

## Referee Comment (RC1) · Anonymous Referee #1 · 14 Nov 2017

The authors develop a model of phytoplankton growth with particular emphasis on dissolved organic matter (DOM) release, based loosely on the Dynamic Energy Budget (DEB) theory. DOM release is described according to the overflow and passive-diffusion hypotheses and formulated as balancing the stoichiometries of several cellular compartments. The model is partially calibrated with a dataset and several potential implications are discussed.

General evaluation

This ms leaves me feeling somewhat lost. To begin with, the model appears overly complex compared to the data used for calibration. Table 2 shows more than 40 pa-

rameters, only slightly less than the number of independent data in Fig. 2. While a few parameters were set to literature values, it still remains difficult to believe that all other parameters could be constrained by the data. This leads to the second major problem, the lack of a proper model validation, which I consider fundamental for any modelling paper. The fact that this applies also to previous publications cited as the foundation for the model further adds to the rather flaky impression of the present ms. The third major problem is in the lack of justification (mechanistic or empirical) or wrong justification of several central model equations. Hence, all conclusions and implications appear unjustified. The last major problem I have with this ms is that I did not find clear statements about the assumptions, goals, and findings of this study, nor does it appear to add anything really new to the topic of DOM production by phytoplankton. In summary, the study design and model formulation appear so fundamentally compromised that I cannot foresee how this ms could be salvaged within a reasonable number of revisions.

Specific points

1. Model complexity and parameter estimation

The authors use one of several datasets published by Flynn et al. (2008) for calibrating most of the model parameters. They do not specify how the calibration was done but only state that they were tuned. They also do not explain why they chose this particular dataset, although many more exist (some of them cited in the ms). The sensitivity analysis is rather cursory and does not consider parameter covariances nor correlations, for which the authors obviously would have had to consider a much larger dataset. However, the sheer number of parameters considered important (Table S1) would overwhelm any parameter-estimation method I know of. Thus, it is not only a problem of the extent of the dataset but equally one of the complexity of the model formulation.

2. Lack of validation

I consider a proper model validation essential to any modelling study. Validation can be, and often is, done together with the model calibration. But this is possible only if the number of estimated parameters is relatively small compared to the number of model variables and also relative to the number of datasets used for calibration. An example of this is the Geider et al. (1998) model (Limnol. Oceanogr.), but it does not apply here. The problem is one of model falsifiability. The large number of sensitive model parameters suggests that this model can be tuned to any kind of data. This means that it has no explanatory power unless the calibrated model is compared to other datasets not used for calibration.

3. Lack of or wrong justification of model equations

Eq. (4) is an unusual form of the light dependence and needs justification. Invoking NADPH as the agent responsible for P limitation is not very logical as NADPH contains only a tiny fraction of cellular P even under severe P limiting conditions. See, e.g., Ågren's (2004) model in Ecology Letters and refs. therein for a discussion of how P limitation works.

Eqs. (5) and (6) are not explained or derived, nor is a ref. given. I have not seen this form of cell-quota dependence before and wonder why this form, with three or four parameters (depending how they are counted) was chosen over Droop's model, which has only two parameters.

Applying Eq. (7) to C assimilation does not appear justified. $CO_2$ enters the cell passively, often enhanced by some form of $CO_2$ concentrating mechanism (CCM) in the chloroplast, so the kinetics are very different from Eq. (7).

I find Eq. (9) problematic for several reasons. First off, it treats the CB cycle as a single enzyme reaction, which is not the case. Secondly, and more importantly, the $CO_2$ concentration inside the chloroplast ist usually kept high enough by the CCM so that RuBisCO is saturated. Thus, the rate of $CO_2$ fixation is mostly controlled by the rate of light harvesting, as represented here by NADPH arrival rate, and the availability of

RuBisCO and the other enzymes of the reaction centers, but not by CO2 concentration or the substrates for NADPH synthesis.

The use of different handling rates for NO3 and NH4 in Eq. (10) does not make sense to me. If DIN (NO3 or NH4) is not limiting the rate-limiting step of N assimilation is protein biosynthesis, as NO3 reduction and formation of amino acids are very fast processes. But the maximum rate of protein biosynthesis should not depend on whether the N derives from NH4 or NO3.

NO3 excretion (Eq. 15) does not make sense to me. Maybe one can consider some leakage but most models treat uptake as the net rate already accounting for leakage. I know that some phytoplankton can reduce NO3 to counter photo-inhibition and then release NH4 but I have never seen a report of phytoplankton excreting NO3.

The authors never provide any details how they implemented their model in Matlab. Did they program their own ODE solver or use one of the built-in functions?

The model appears to rest on many implicit, i.e., unspecified, assumptions. Some of the most disconcerting ones concern the internal reserves for NO3 and NH4. While phytoplankton may generally have the ability to store N in the form of amino acids or oligo- or polypeptides etc., storage of inorganic N has been shown only for very few species, and these are very large and very rare.

While none of these problems are directly related to DOM release, it is clear that the model does not rest on firm grounds. Thus, any calculations regarding DOM release are pointless because they lack a sound underpinning.

4. Lack of clarity, goals, etc.

With respect to the patterns of DOM release and its relation to primary production, it is now generally accepted that these result from interactions among all plankton groups, i.e., bacteria, phytoplankton, zooplankton, detritus, etc. Phytoplankton DOM release is only one of these and while it may well be the dominant source of DOC,

whether this applies to DON and DOP as well is still an open question. I think one cannot understand the relation between DOM release and primary production without reference to other processes in plankton ecosystems. But these are missing in the discussion.

In their conclusions, the authors write that their model could be coupled with biogeo-chemical models (on what scale? 1D, local, global?) but they do not provide any arguments for this anywhere else in the ms. I think a coupling of such a complex model, even if it were theoretically sound, to biogeochemical models is not feasible, simply because it is impossible to constrain its many parameters. Other conclusions are mentioned in the discussion but they appear very speculative. For example, the association of the DOM release mechanism (passive or overflow) with refractory and labile DOM is not explained mechanistically. The way these conclusions are presented, they could equally have been derived directly from observations. It remains unclear how the model relates to them. The point of developing a model, at least as I see it, is to provide insight into how certain mechanisms work, e.g., here it could have been to explain why the overflow mechanism produces more refractory DOC, but this has not been done here. While I do recognise that, technically, the specific model setup is new, I do not see any new insight in terms of mechanistic understanding of DOM release from phytoplankton in this ms.

---

## Referee Comment (RC2) · Anonymous Referee #2 · 16 Nov 2017

This work presents a DEB-based model purporting to describe the production of DOM by phytoplankton under N or P stress. The general topic is important, and does align with the subject areas of the journal.

The model is shown to broadly align with a single data set for N-limitation, and is then operated under different conditions with an aim to consider differential consequences of N vs P limitation. I have the following general observations upon this work which I am afraid makes me question the usefulness of the approach and application.

Firstly, the structure of the model, as shown in Fig.1, is contrary to that which aligns with the physiology of real phytoplankton. Indeed the structure, with its partitioning of

inorganic N and P and within different organic structures appears most strange. I find this very worrying. The model is complex, and claims to be mechanistic (as are DEB models), but it just does not tally.

There are also other facets of the model description that are of concern to me: the stoichiometry of cellular components (which is not shown to collectively reproducing observed changes in cellular C:N:P); the explanation of internal ammonium (which in reality is essentially zero) and of internal nitrate (which is also usually very low, and indeed contrary to some reports can never attain a significant % of cell-N because of nitrate-solubility issues) and the interactions between ammonium and nitrate usage (which perpetuate various classic misleading literature articles); the whole rational for DIN and DIP release (perhaps a hangover from the heterotrophic origins of the DEB concept?) appears to be unsupported unless the simulations are running in a light-dark cycle (are they?); comments about cell size variation gloss over the fact that P-limited cells are very much larger than nutrient-replete cells, and N-limited cells are much smaller; and so on.

I therefore have a serious problem with the conceptual basis of the model. The authors have also not explained why their approach has any benefits over any other approach.

The model is shown to fit against only one data set, for a diatom, growing under N-limitation. That data set comes from one of those presented in Flynn et al. 2008, who also conducted a (solely N-based) exploration of the description of DOM release. Very strangely there is no comparison with the utility of this model with that of Flynn et al for N-based scenarios; why have the authors not done this?

Further, while this article considers P-limitation (Flynn et al did not) it does so using a framework that is unproven, and hence one that must remain speculative. Indeed, no evidence is presented for how the model handles P-limitation, or indeed how cellular C:N:P varies under N and P limitation. If the DEB model had at least been shown to handle general C-N-P interactions (there are various data sets available to which to

conduct such a comparison) then this would not be of such concern.

The model is sold for potential deployment in ecosystem simulators but it appears incomplete for such usage, lacking acclimative Chl:C (which is important in DOM modelling as a failure to rapidly modulate C-fixation promotes DOM release) and indeed it has not been shown how the model reacts to light-limitation (which is of importance during bloom development and thence to DOM release).

In section 4.2 there is a commentary about P vs N –limitation; I suggest this requires some common basis for reference (perhaps u/Umax?). As it stands the statements appear ambiguous and potentially incorrect.

In section 4.3 is a discussion about forms of DOM. I find the description of DOC, DON, DOP used in this article somewhat confusing; DON and DOP are also components (subsets) of DOC. The discussion lacks a consideration of CNP of DOM forms, and also (again) a comparison with the outputs of the Flynn et al. effort. Just now it is not clear to me what advantages this DEB-based approach may have over any other model. There are also some strange (to me) comments concerning glucose and polysaccharides in this section.

There are various detailed comments that I could give to help the authors, but just now I think that I need to see:

i) a more acceptable conceptual basis (I do not believe that Fig.1 does this),

ii) a demonstration that the whole model can describe dynamic C:N:P experimental data series,

iii) a more rigorous set of comparisons with Flynn et al (whose data they use, but then for some reason never further discuss in comparative terms even in the context of N-limited growth) and indeed with the functionality of other models (ERSEM springs to mind).

---

## Author Comment (AC1) · 17 Dec 2017

Dear Editor,

Please consider below our responses to first referee's comments to our manuscript, entitled "Dissolved organic matter release by phytoplankton in the context of the Dynamic Energy Budget theory", by E. Livanou, A. Lagaria, S. Psarra and K. Lika. We thank referee 1 for the time invested and for the thorough review of our manuscript. In the following revision notes we give detailed responses to his comments and provide relevant manuscript modifications that could be applied in the ms for further clarification (referee's comments are presented in bold, our replies in blue and relevant changes in the initial text are highlighted in red to facilitate reading). All citations in our responses are listed at the end of the document. Citations in the initial and modified text are included in the references list of the original ms.

**Referee #1**
**The authors develop a model of phytoplankton growth with particular emphasis on dissolved organic matter (DOM) release, based loosely on the Dynamic Energy Budget (DEB) theory. DOM release is described according to the overflow and passive diffusion hypotheses and formulated as balancing the stoichiometries of several cellular compartments. The model is partially calibrated with a dataset and several potential implications are discussed.**

**General evaluation**
**This ms leaves me feeling somewhat lost. To begin with, the model appears overly complex compared to the data used for calibration. Table 2 shows more than 40 parameters, only slightly less than the number of independent data in Fig. 2. While a few parameters were set to literature values, it still remains difficult to believe that all other parameters could be constrained by the data. This leads to the second major problem, the lack of a proper model validation, which I consider fundamental for any modelling paper. The fact that this applies also to previous publications cited as the foundation for the model further adds to the rather flaky impression of the present ms. The third major problem is in the lack of justification (mechanistic or empirical) or wrong justification of several central model equations. Hence, all conclusions and implications appear unjustified. The last major problem I have with this ms is that I did not find clear statements about the assumptions, goals, and findings of this study, nor does it appear to add anything really new to the topic of DOM production by phytoplankton. In summary, the study design and model formulation appear so fundamentally compromised that I cannot foresee how this ms could be salvaged within a reasonable number of revisions.**

**Response**
In this ms we propose a model for phytoplankton growth with focus on dissolved organic matter (DOM) release. Our aims, clearly stated in the end of the Introduction section (p. 3 lines 11-14) are to: "*1) investigate the effects of N- and P-limitation on DOM release by phytoplankton, 2) resolve the relationship between primary production and DOM release and 3) elucidate the mechanisms under the two conceptual processes of DOM production, namely, passive diffusion and active exudation*".
The novelty of the model lies in its ability to capture the two discrete physiological mechanisms of DOM production. It does so without making further assumptions but rather via the existence of two alternative pathways that emerge from the theory. To the best of our knowledge this is the first attempt to model mechanistically the active exudation that reflects an unbalanced availability of nutrients.

The major findings are summarized in the Conclusions section (p.17, lines 2-11: "*the distinction and assessment […] prevalence of the active exudation mechanism*") and also briefly in the abstract.

We do acknowledge that the resulting model is quite complex. However, how much complexity may be required depends on what processes the model tries to capture. It is well known that more than one process contribute to DOM release and as Flynn et al. (2008) demonstrated "net DOM release should be modeled as functions of microalgal nutritional status and growth rate". In the same study the authors compared models with different levels of complexity and arrive at the conclusion that the simple ones may fit well the experimental data but for wrong reasons. Justification of model complexity is also discussed in details below (see response to specific point 1).

The issues raised by referee 1 regarding the validation of the model and the justification of its structure are also discussed in details below (see response to specific points 2 and 3).

Overall, the proposed model for phytoplankton is fully based on multiple reserves DEB model (chapter 5, Kooijman (2010)). So we strongly disagree with the comment of referee 1 that the model is "based loosely on DEB theory".

**Referee #1**
**Specific points**
**1. Model complexity and parameter estimation**
**The authors use one of several datasets published by Flynn et al. (2008) for calibrating most of the model parameters. They do not specify how the calibration was done but only state that they were tuned. They also do not explain why they chose this particular dataset, although many more exist (some of them cited in the ms). The sensitivity analysis is rather cursory and does not consider parameter covariances nor correlations, for which the authors obviously would have had to consider a much larger dataset. However, the sheer number of parameters considered important (Table S1) would overwhelm any parameter-estimation method I know of. Thus, it is not only a problem of the extent of the dataset but equally one of the complexity of the model formulation.**

**Response**
The ideal data set for a DOM release model should derive from an axenic phytoplankton culture demonstrating mass balance for all elements considered in the model. For our work, the most suitable data set that we were able to find in the literature was that of Flynn et al. (2008). This study had excellent data sets from ammonium and nitrate growing batch cultures, for a variety of phytoplankton species. However, we couldn't find data for any species that would simultaneously describe carbon, nitrogen and phosphorus dynamics in both the organic and inorganic particulate and dissolved pools. *Thalassiosira pseudonana* was chosen over the other species because for this species we were also able to find supplementary information regarding its molar elemental ratios in terms of C:N:P under nitrogen and phosphorus limited growth (Perry, 1976; Goldman et al., 1979, Flynn et al., 2008). Therefore, as stated in the methods Sect. 2.3 (p.9 lines 26-27), in addition to the data set of Flynn et al. (2008), the molecular elemental ratios of biomass (i.e. C:P, C:N, N:P) were also used in order to constrain the parameter values so that the resulting ratios (Fig. 8a-c in the ms) would fall close to observable ranges reported in Perry (1976), Goldman et al. (1979) and Flynn et al. (2008). We intend to explicitly state that in the revised manuscript by modifying the text on Sect. 2.3 (p.9, lines 26-27) as follows:

*"Furthermore, the molecular elemental ratios of biomass (i.e. C:P, C:N, N:P) were also used in order to constrain the parameter values so that the resulting ratios would fall close to observable ranges reported in Perry (1976), Goldman et al. (1979) and Flynn et al. (2008)."*

We recognize that the model is complex and parameter rich, however, an adequate description of DOM excretion would have not been possible if variable biomass stoichiometry and, thus, multiple

reserves were not taken into account. In addition, the need to consider multiple reserves stems from the strong homeostasis assumption of DEB theory which implies that when nutrient assimilation pathways are independent, as is the case for phytoplankton, multiple reserves should be taken into account, one for each assimilation pathway (Kooijman, 2010).

Regarding the sensitivity of the model, the aim of the analysis was to identify parameters that affect the model outputs of interest, and whether they remain influential at all phases of growth. We show that DOM release and biomass, the two model outputs of interest, are sensitive to the parameters that are related to photosynthesis light and dark reactions and to stoichiometric coefficients coupled to carbohydrates reserves (as it would be expected).

**Referee #1**
**2. Lack of validation**
**I consider a proper model validation essential to any modelling study. Validation can be, and often is, done together with the model calibration. But this is possible only if the number of estimated parameters is relatively small compared to the number of model variables and also relative to the number of datasets used for calibration. An example of this is the Geider et al. (1998) model (Limnol. Oceanogr.), but it does not apply here. The problem is one of model falsifiability. The large number of sensitive model parameters suggests that this model can be tuned to any kind of data. This means that it has no explanatory power unless the calibrated model is compared to other datasets not used for calibration.**

**Response**
We agree with the referee that model validation is essential. In our case, due to insufficient experimental data on DOM release by phytoplankton, we relied mainly on qualitative rather than quantitative match between model predictions and experimental evidence. The model parameters' values used in the simulations were not arbitrary, but they were either obtained from the literature or were constraint-based values. Our goal in this study was not to make quantitative predictions of DOM release rate, but rather using a logical parameter set to perform a theoretical exercise (i.e., growth of phytoplankton under different nutrient availability scenarios) in order to better understand the mechanisms of DOM production by phytoplankton. However, although our approach is qualitative it still results in highlighting the nature of the mechanisms for DOM release.

**Referee #1**
**3. Lack of or wrong justification of model equations**
**Eq. (4) is an unusual form of the light dependence and needs justification. Invoking NADPH as the agent responsible for P limitation is not very logical as NADPH contains only a tiny fraction of cellular P even under severe P limiting conditions. See, e.g., Ågren's (2004) model in Ecology Letters and refs. therein for a discussion of how P limitation works.**

**Response**
Eq. (4) is a hyperbolic function of light intensity, the well known Monod-Michalies-Menten (M-M-M) type, with some modifications. The derivation of Eq. (4) in this form is based on the concept of synthesizing units of DEB theory (Kooijman 1998, 2010) and it is derived elsewhere (e.g.,Poggiale et al. (2010), thus, we did not think it was necessary to include it in our ms. However, we intend to include more details on the derivation of Eq. (4) in the supporting information of the revised ms. The modifications in M-M-M equation include the dependence of photon's binding probability and NADPH formation, respectively, on N and P content of the cell through the formulations $v_p$ and $\rho_L$.

In our model we do not consider NADPH as the agent for P-limitation but rather that P-limitation could have an adverse effect on carbohydrate production. We model photosynthesis in two steps.

First, photons are bound by the photosynthetic units (PSUs) that produce NADPH. Then, the NADPH and the carbon dioxide are bound by the $SU_{CB}$ to deliver the product of the transformation, namely, carbohydrates. There is evidence that steps in this process are affected under P-limitation. Evidence from higher plants suggests that P stress affects the carboxylation capacity (de Groot et al., 2003). Moreover, Geider et al. (1993) proposed that RuBisCO activity may be impaired due to insufficient regeneration of ribulose bis-phosphate, which in turn is related to reduced supplies of ATP and NADPH, as a result of phosphorus deficiency. In addition, under P-limiting conditions, a slow-down of photosynthesis in soybean plants has been observed (Fredeen et al., 1990). The authors suggest that, although ATP and NADPH levels decreased by 40-50% relative to control, the agent responsible for this reduction was probably the regeneration of ribulose bis-phosphate by reduced Calvin-cycle enzymes activity.

Given the available experimental evidence we used the empirical Eq. (5) to reduce carbohydrate production via reduction of the NADPH production (Eq. 4). Alternatively we could model it via Eq. (9).

To further support our choice we intend to add the citations of Fredeen et al. (1990) and de Groot et al. (2003) in the revised ms.

Ågren (2004)is an interesting study proposed by referee 1 and we thank him for pointing it out. Ågren (2004) recommended by the referee as explanatory paper on P-limitation, suggests that P is the main element of ribosomes, while the additional P in the cells will be proportional to the amount of C. Thus, low P-content has a direct effect on growth rate. In DEB theory and, subsequently, in our model, the effect of P availability on growth rate is taken into account through the formation of generalized reserves E, E′ (Eq. 10) where carbon, nitrogen and phosphorus are essential in a constant stoichiometry to produce the generalized reserves, which can contain various macromolecules such as RNA, proteins, lipids etc. Generalized reserves will subsequently give the catabolic flux in order to cover maintenance costs and increase the structural mass.

**Referee #1**
**Eqs. (5) and (6) are not explained or derived, nor is a ref. given. I have not seen this form of cell-quota dependence before and wonder why this form, with three or four parameters (depending how they are counted) was chosen over Droop's model, which has only two parameters.**

**Response**
The function in Droop's model is hyperbolic resulting in rapid decrease to zero as cell-quota approaches the minimum value. This means that, close to minimum cell-quota values, small changes in cell-quota will result in large decrease in photosynthesis (in our case). Experimental evidence suggests that under nutrient limited conditions photosynthesis, although reduced, is sustained at positive values (e.g. Granum et al., 2002; Kamalanathan et al., 2016). Thus, we were seeking for a function that would allow a smooth decrease from a max to min value. Sigmoid functions are good candidates. The use of sigmoid functions is not new in phytoplankton models (see for example, Bonachela et al. (2013), Flynn et al. (1997). One way to mathematically describe a sigmoid function is as given in Eqs. (5) and (6). The parameters $b_P$ and $b_N$ control the shape of the function. Other mathematical functions could be used without altering the qualitative behaviour of the model. The references of Flynn et al. (1997), Bonachela et al. (2013) will be included in the revised ms.

**Referee #1**
**Applying Eq. (7) to C assimilation does not appear justified. CO2 enters the cell passively, often enhanced by some form of CO2 concentrating mechanism (CCM) in the chloroplast, so the kinetics are very different from Eq. (7).**

**Response**

Several studies have shown that both $CO_2$ and $HCO_3^-$ are taken up by marine eukaryotic microalgae (Burkhardt et al., 2001; Tortell et al., 2006). $CO_2$ are small, uncharged molecules that can pass the membrane passively (Neven et al., 2011) but also numerous pieces of evidence exist of active $CO_2$ transport in cyanobacteria ( Miller et al., 1988; Espie et al., 1991), diatoms and green algae (Rotatore and Colman, 1992). $HCO_3^-$ ions are charged and, thus, can be taken up either by active transport, by transporters in the cell membrane, or indirectly by transformation of $HCO_3^-$ to $CO_2$ in the cell boundary layer by extracellular carbonic anhydrases ( Sültemeyer et al., 1993; Giordano et al., 2005). Furthermore, simultaneous uptake of $CO_2$ and $HCO_3^-$ has been observed by marine diatoms and the uptake of inorganic carbon ($CO_2$ and $HCO_3^-$ ) is usually described by Michaelis-Menten kinetics (Burkhardt et al., 2001; Rost et al., 2003). Thus, based on the literature discussed here we modelled the uptake of inorganic carbon by the phytoplankton cell using the Eq. (7). Supplementary information will be included in the revised ms.

**Referee #1**

**I find Eq. (9) problematic for several reasons. First off, it treats the CB cycle as a single enzyme reaction, which is not the case. Secondly, and more importantly, the CO2 concentration inside the chloroplast is usually kept high enough by the CCM so that RuBisCO is saturated. Thus, the rate of CO2 fixation is mostly controlled by the rate of light harvesting, as represented here by NADPH arrival rate, and the availability of RuBisCO and the other enzymes of the reaction centers, but not by CO2 concentration or the substrates for NADPH synthesis.**

**Response**

Although Calvin-Benson cycle is not a single enzyme reaction, our approach is based on the Synthesizing Unit concept which corresponds to a generalized enzyme or a complex of enzymes that binds one or more substrate molecules to deliver a product molecule. For modeling purposes, we assume that the Calvin Benson SU ($SU_{CB}$) corresponds to a complex of enzymes that binds and processes, in a parallel transformation, NADPH and $CO_2$ molecules and delivers carbohydrates as the product of this transformation. As it is stated in the ms (Sect. 2.1, p.3, line 25) readers can refer to Kooijman (1998, 2010) for a detailed description of the assumptions and kinetics of the SU, while a detailed description of the derivation of Eq. (9) is given in Lika and Papadakis (2009) also cited in the ms (Sect. 2.3, p.3, line 27 and Sect. 2.2.1, p.6, lines 24-26).

$CO_2$ is a required compound for the RuBisCO. The quantification of $CO_2$ fixation through Eq. (9) allows for a dynamic and direct link with the inorganic carbon in the medium. Note, however, that when $CO_2$ concentration is high Eq. (9) is reduced to a hyperbolic equation of NADPH concentration and, thus, to light harvesting.

**Referee #1**

**The use of different handling rates for NO3 and NH4 in Eq. (10) does not make sense to me. If DIN (NO3 or NH4) is not limiting the rate-limiting step of N assimilation is protein biosynthesis, as NO3 reduction and formation of amino acids are very fast processes. But the maximum rate of protein biosynthesis should not depend on whether the N derives from NH4 or NO3.**

**Response**

Indeed, protein biosynthesis does not depend on whether N is derived from $NO_3$ and $NH_4$ and we account for this assuming the same values for $k_{NH}$ and $k_{NO}$ (Table 2 in the ms). However, we prefer to give a general formulation for the process.

Furthermore we modified text in Sect. 2.2.2 (p. 7, lines 17-19) as follows:

"$j_{E,A_m}$ denotes the structure-specific maximum assimilation rate of generalized reserves which is not necessarily constant: $j_{E,A_m} = \theta_{NO}^A/k_{NO} + \theta_{NH}^A/k_{NH} \cdot k_{NO}$, $k_{NH}$ are, respectively, the handling rates of nitrate and ammonium, which can be assumed to have the same value, given the fast reduction of nitrate".

**Referee #1**
**NO3 excretion (Eq. 15) does not make sense to me. Maybe one can consider some leakage but most models treat uptake as the net rate already accounting for leakage. I know that some phytoplankton can reduce NO3 to counter photo-inhibition and then release NH4 but I have never seen a report of phytoplankton excreting NO3.**

**Response**
As the referee mentions there aren't any reports of $NO_3$ excretion by phytoplankton. However, the rationale behind Eq. 15 is thoroughly justified in the Supplementary Information (Sect. S2, p.2, lines 6-19). In the ms, in line 29, p.8 we refer the reader to the Supplementary Information for more details on Eq. (15). For clarity we also subjoin here the relevant paragraph from the supplementary information

"It has been found that phytoplankton release part of the assimilated nitrate in the form of nitrite (Parslow et al., 1984) which can be up to 50 % of the assimilated nitrate (Collos, 1998; Lomas et al., 2000). It seems that nitrite release is widespread in marine phytoplankton. It can be considered as an active exudation process that links to the nitrate uptake (Collos, 1998) and allows phytoplankton to avoid excessive nitrite intracellular concentration (Malerba et al., 2012). Excretion of nitrite has been observed when phytoplankton is replete with nitrate but experiences low-light availability conditions (Kiefer et al., 1976; Flynn and Flynn, 1998; Mordy et al., 2010; Shriwastav et al., 2014). This may result in decoupling of the assimilatory pathways as the relative activity of nitrite reductase to nitrate reductase is reduced (Sciandra and Amara, 1994; Lomas and Lipschultz, 2006; Mordy et al., 2010). This is due to the fact that nitrite reductase requires the light depended ferredoxin as the electron donor, which is synthesized only during photosynthesis (Collos, 1998). In addition, if there are enough carbon skeletons available the excess nitrogen may be exuded in an organic form therefore contributing to the DON pool (Lomas et al., 2000). Here, we hypothesize that once the nitrate reserve is mobilized, if is not further reduced to ammonium and channelled through the catabolic flux, due to stoichiometric constraints, it will be rejected by the $SU_2$ and excreted into the surrounding medium either as nitrite, which in the model is implicitly added to the NO or as DON which is added to the DON pool, while part of it will be reincorporated in the $E_{NO}$ reserve".

**Referee #1**
**The authors never provide any details how they implemented their model in Matlab. Did they program their own ODE solver or use one of the built-in functions?**

**Response**
We omitted to mention that we used the built-in function ode45 in Matlab. This will be added in the revised manuscript.

**Referee #1**
**The model appears to rest on many implicit, i.e., unspecified, assumptions. Some of the most disconcerting ones concern the internal reserves for NO3 and NH4. While phytoplankton may generally have the ability to store N in the form of amino acids or oligo- or polypeptides etc., storage of inorganic N has been shown only for very few species, and these are very large and very**

**rare. While none of these problems are directly related to DOM release, it is clear that the model does not rest on firm grounds. Thus, any calculations regarding DOM release are pointless because they lack a sound underpinning.**

**Response**

We disagree with the statement of referee 1 regarding the internal reserves of $NO_3$ and $NH_4$. A substantial body of literature has confirmed that inorganic nitrogen pools can be accumulated in laboratory cultures of representative marine diatoms (e.g. *Skeletonema costatum, Thalassiosira weissflogii, Thalassiosira pseudonana*) and flagellates (e.g. *Dunaliella tertiolecta, Pavlova lutheri, Isochrysis galbana*) species (Dortch et al., 1984; Lomas and Glibert, 2000; Lourenço et al., 1998; Raimbault and Mingazzini, 1987). In addition, the ability of phytoplankton to store inorganic nitrogen (mainly nitrate) has been shown also in natural phytoplankton assemblages ( Dortch et al., 1985; Pettersson, 1991; Bode et al., 1997). Based on these studies, we include in our model the reserves for nitrate and ammonium in order to simulate the well known behaviour of excess uptake and internal accumulation of inorganic nitrogen when uptake rate exceeds growth rates. This approach is not new, as internal pools of inorganic nitrogen (i.e. reserves) are also explicitly modelled in other phytoplankton models (Flynn et al., 1997; Talmy et al., 2014; Ghyoot et al., 2017). Furthermore, DEB theory's strong homeostasis assumption (i.e. the chemical composition of a reserve or a structure does not change during the organism's life cycle) implies that multiple reserves should be considered, one for each assimilation pathway, when they are independent as is the case for phytoplankton.

However, it is indeed acknowledged in the literature that there are significant differences among species in their ability to accumulate large intracellular nitrate or ammonium pools, with larger species having larger pools (Dortch et al., 1984) as referee 1 points out. Our aim was to present a generalized model for phytoplankton. Differences among species regarding the extent to which ammonium and/or nitrate are accumulating intracellularly can be accommodated by the model via the choice of parameter values. For example, if for a certain species intracellular ammonium pools do not accumulate a very large turnover rate for ammonia reserve would give an extremely low reserve density for ammonia and, thus, the ammonia reserve could be omitted (Kooijman 2010, chapter 5).

In addition, as stated by referee 1, phytoplankton stores N in the form of amino acids, proteins and nucleic acids provided that there are sufficient carbon skeletons for the synthesis of these compounds. In our model we do not model explicitly the amino acids pool, however we made the following assumption stated in Sect.2.2.2 (p.7, lines 26-29) and here for clarity:

*"Given that the incorporation of ammonium into organic molecules is a fast process (Stolte and Riegman, 1995) we assume that the $E_{NH}$ reserve contains a mixture of ammonium and DON fraction of free amino acids. Accordingly, we assume that the DOC fraction of the free amino acids is contained in the $E_{CH}$ reserves."*

Regarding proteins and nucleic acids that can also function as reserves of nitrogen, we assume that, since the synthesis of these compounds involves many steps and in most cases, requires also the incorporation of phosphorus (e.g. RNA phosphorylated proteins), these compounds are collectively represented by the generalized reserve E which is assumed to be a mixture of chemical compounds that does not change in composition and thus have a fixed stoichiometry. In order to clarify better the concept of generalised reserves we will modify text in Sect. 2.1 (p.3, lines 31-32) as follows:

*"[…] to form the generalized reserves (E), which consist of a mixture of chemical compounds (RNA, proteins, lipids etc) and they have a fixed stoichiometry."*

**Referee #1**
**4. Lack of clarity, goals, etc.**
**With respect to the patterns of DOM release and its relation to primary production, it is now generally accepted that these result from interactions among all plankton groups, i.e., bacteria, phytoplankton, zooplankton, detritus, etc. Phytoplankton DOM release is only one of these and while it may well be the dominant source of DOC, whether this applies to DON and DOP as well is still an open question. I think one cannot understand the relation between DOM release and primary production without reference to other processes in plankton ecosystems. But these are missing in the discussion.**

**Response**
Referee 1 validly points out those other sources and processes that may contribute to the release and fate of DOM. However, our focus in this paper was on the direct DOM production by the phytoplankton cells and not on processes that may result in physical disruption of the cells such as sloppy feeding or viral lysis. In order to acknowledge that other sources/processes may be related to the observed patterns of DOM release and its relation to primary production in the food web context we added the following lines in the Discussion (Sect. 4.3, end of last paragraph).

*"However, from an ecosystem point of view, other food web processes of indirect release of DOC by phytoplankton, such as sloppy feeding by zooplankton and viral lysis, may also contribute to the patterns between primary production and DOC release and bacterial production, observed in oligotrophic environments (Teira et al., 2001)."*

And in the Conclusions (p.17) we modify lines 9-11 as follows:
*"Our model, by allowing to discriminate between the two mechanisms of DOM production, suggests that the decoupling of primary production and DOC release and of primary production and bacterial production, often observed in oligotrophic conditions may be related to the prevalence of the active exudation mechanism."*

**Referee #1**
**In their conclusions, the authors write that their model could be coupled with biogeochemical models (on what scale? 1D, local, global?) but they do not provide any arguments for this anywhere else in the ms. I think a coupling of such a complex model, even if it were theoretically sound, to biogeochemical models is not feasible, simply because it is impossible to constrain its many parameters.**

**Response**
The model structure allows for a quantification of the production of the distinct size fractions of phytoplankton exudates produced by the two mechanisms of DOC release (passive diffusion, active exudation). Based on experimental evidence, we assume that these different size fractions, originating from different cellular processes, will have distinct molecular composition and thus different availability properties for bacteria. Furthermore the model allows for keeping track the stoichiometry of DOM produced, which also affects the degradability of DOM. Therefore, we suggested that the proposed model could be used, potentially, in ecosystem simulations in order to resolve the degradability of DOM produced by phytoplankton.
Other potentials of this model formulation is to include acclimated Chla:C (also important in ecosystem simulations), for example through light dependence of the parameter $\rho_L$ (Papadakis et al., 2012). This will require further analysis and it is beyond the scope of this ms. Also the scale at which a model operates depends on its purpose.

**Referee #1**
**Other conclusions are mentioned in the discussion but they appear very speculative. For example, the association of the DOM release mechanism (passive or overflow) with refractory and labile DOM is not explained mechanistically. The way these conclusions are presented, they could equally have been derived directly from observations. It remains unclear how the model relates to them. The point of developing a model, at least as I see it, is to provide insight into how certain mechanisms work, e.g., here it could have been to explain why the overflow mechanism produces more refractory DOC, but this has not been done here. While I do recognise that, technically, the specific model setup is new, I do not see any new insight in terms of mechanistic understanding of DOM release from phytoplankton in this ms.**

**Response**
The novelty of the model lies in its ability to capture the two physiological mechanisms of DOM production. It does so without making further assumptions but rather via the existence of two alternative pathways that emerge from the theory.

The production of refractory or labile DOC is not modelled explicitly and it would have been impossible to do so based on our current understanding of the physiology of DOM release by phytoplankton (Thornton, 2014). While there is no method to discriminate and measure experimentally the rates of DOC production via the two processes, the relative contribution of these processes can be indirectly estimated based on the size classes of the produced DOC (Thornton, 2014)

In our model, we associate passive leakage to growth and death process of DOC release and active exudation to rejection of unprocessed substrates by the SU. We based this link on the findings from various studies looking into either the molecular composition of DOC during the different states of the cultures (exponentially, stationary) (e.g., Urbani et al., 2005) or the molecular composition of high and low molecular weight DOC (e.g., Borchard and Engel, 2015). In order to make model assumptions clearer regarding the relative presence of glucose and heteropolysaccharides of various monemer composition of excreted DOC, based on both referee's comments, we intend to modify Sect.3.2.3 (p. 13, lines 1-10) *"In order to further investigate [...] exuded by the cell due to unbalanced growth (Fogg, 1983; Urbani et al., 2005; Flynn et al., 2008; Borchard and Engel, 2015)"* in the revised ms as follows:

*"Borchard and Engel (2015) showed that in steady-state, P-limited cultures of Emiliania huxleyi glucose was the dominant monomer in both the small size fraction (1–10 kDa) of dissolved polysacchrides and in the particulate fraction (cell content) and less significant in the larger size fractions (>10 kDa), which contained a variety of monomers. They suggested that, due to their size and resemblance to the cellular material, low molecular weight carbohydrates should be released by passive diffusion. On the other hand, high molecular weight carbohydrates, due to their size and distinct composition from the cellular material, should be produced via active exudation (Borchard and Engel, 2015). Furthermore, in cultures of marine diatoms, glucose has been identified as the most abundant monomer during the exponential, nutrient-replete phase of growth in the extracellular carbohydrates. A pronounced decrease of glucose and an increase of heteropolysacharides, containing various monomers, has been observed in the stationary, nutrient-limited phase (Underwood et al., 2004; Urbani et al., 2005). Based on these evidence we define as $DOC_L$ the DOC produced as a result of the fluxes associated with growth ($j_{DOC,G}$ , Fig. 6, dashed line) and death ($j_{DOC,D}$, Fig. 6, dash-dot line) processes and we relate these two fluxes to the mechanism of passive diffusion mechanism. Thus, $DOC_L$ should be small in size and contain mono- and polysacharides, rich in glucose, and also DOC associated with nitrogen or phosphorus containing*

*compounds that can be released from exponentially growing cells or from cell lysis. On the other hand, we define as $DOC_H$ the DOC produced as a result of the $j_{DOC,R}$ flux (Fig. 6, solid line) that corresponds to the rejection flux of unprocessed substrates by the SU. Consequently, this flux is related to the mechanism of active exudation. $DOC_H$ should contain high molecular weight (>10kDa) heteropolysaccharides, poor in glucose and also DOC associated with nitrogen or phosphorus containing compounds that could be exuded by the cell due to unbalanced growth."*

And in the Discussion, after also considering referee's 2 comments, we will replace text in Sect.4.3 (p.15, lines 20-34 and p.16, 1-9) *"In this study, using the DEB model […]consisting of a variety of monomers, have been found to escape bacterial degradation (Obernosterer and Herndl, 1995; Hama and Yanagi, 2001; Puddu et al., 2003)"* as follows:

*"In this study, using the DEB model for phytoplankton (Kooijman, 2010), we were able to discriminate between the two major conceptual mechanisms of DOM release, i.e., passive diffusion and active exudation, in contrast to existing phytoplankton models that involve DOM exudation (Van Den Meersche et al., 2004; Schartau et al., 2007; Flynn et al., 2008; Kreus et al., 2014) but do not discriminate between the mechanisms of DOM release. For example, the most complex model presented in Flynn et al. (2008) employed an empirical description that related the relative rate of leakage of DOC and DON to the N:C status of the cells. On the contrary, in our modeling approach, the theory of SU quantifies the active exudation of the non-limiting compounds. Since the nutrients are taken up independently, one or more catabolic fluxes can limit the synthesis of the generalized reserves $E'$. In that case, the non limiting "molecules" will occupy the binding sites of the $SU_2$ but they will not be processed further due to the absence of the limiting flux and, thus, they will be rejected by the $SU_2$. Subsequently, a fraction of this rejection flux will be excreted. In that way, the effect of nutrient limitation on exudation rate is accounted for. Furthermore, Flynn et al. (2008) assumed a higher rate of leakage until the external concentration attained a critical value in order to account for the rapid accumulation of DOM during the initial stages of the culture. On the other hand, in our model, based on DEB theory's assumptions for product formation, we describe a second process of DOM excretion which is stoichiometrically coupled to the growth rate and results in high rates of DOM production during the initial nutrient-replete phase of the culture. This can be seen as an overhead for growth as this material is passively leaked outside the cell. Thus, the advantage of our approach lies in its ability to capture the two physiological mechanisms of DOM production. It does so without making further assumptions but rather via the existence of two alternative pathways that emerge from the theory.*

*Based on experimental evidence we assumed that the different processes contributing to DOC release produce two distinct types of DOC (i.e., $DOC_L$, $DOC_H$). $DOC_L$, which is related to growth and death processes and thus to the passive diffusion mechanism, is expected to contain low molecular weight carbohydrates that have a similar composition as the cellular material with high content of glucose (Borchard and Engel, 2015), while $DOC_H$ which is related to the rejection flux of unprocessed substrates by the SU and the active exudation mechanism, has a more distinct composition from the cellular material and is rich in high molecular weight heteropolysaccharides, consisting of a variety of monomers (Biersmith and Benner, 1998; Borchard and Engel, 2015). As such, the model suggests that the relative importance of the two mechanisms and, thus, the relative presence of high and low molecular weight carbohydrates with different molecular composition signatures, is dependent on the nutrient status of the cells. Our approach is different of that of Flynn et al (2008) since they did not distinguish between the two mechanisms of DOM production. Thus, in their model they take into account only leakage, which is related to the nutrient status of the cells and produce low molecular weight DOC such as mono and disaccharides and DOC associated with amino acids. High molecular weight DOC is associated with proteins and nucleic acids and it is produced only via cell lysis, that is enhanced under suboptimal growth conditions, while cell lysis will also result in the leakage of low molecular weight DOM stored intracellularly (Flynn et al., 2008).*

*The molecular composition of DOC released may have implications for its subsequent utilization by bacteria. Many studies have shown that exudates, rich in glucose, are taken up rapidly by bacteria while heteropolysaccharides, consisting of a variety of monomers, have been found to escape bacterial degradation (Obernosterer and Herndl, 1995; Hama and Yanagi, 2001; Puddu et al., 2003). Thus, the novelty of our model is that it allows the quantification of the production fluxes associated with the two classes of DOC ($DOC_L$, $DOC_H$) that will contain carbohydrates with different molecular composition signatures and thus, different degree of degradability by bacteria. Furthermore, our model setup allows for the tracking of the elemental composition of photosynthetically produced DOM, which is also important information for the degradability of DOM."*

**References**

Ågren, G. I.: The C:N:P stoichiometry of autotrophs - Theory and observations, Ecol. Lett., 7(3), 185–191, doi:10.1111/j.1461-0248.2004.00567.x, 2004.

Bode, A., Botas, J. A. and Fernández, E.: Nitrate storage by phytoplankton in a coastal upwelling environment, Mar. Biol., 129(3), 399–406, doi:10.1007/s002270050180, 1997.

Bonachela, J. A., Allison, S. D., Martiny, A. C. and Levin, S. A.: A model for variable phytoplankton stoichiometry based on cell protein regulation, Biogeosciences, 10(6), 4341–4356, doi:10.5194/bg-10-4341-2013, 2013.

Borchard, C. and Engel, A.: Size-fractionated dissolved primary production and carbohydrate composition of the coccolithophore Emiliania huxleyi, Biogeosciences, 12(4), 1271–1284, doi:10.5194/bg-12-1271-2015, 2015.

Burkhardt, S., Amoroso, G., Riebesell, U. and Sültemeyer, D.: $CO_2$ and $HCO_3^-$ uptake in marine diatoms acclimated to different $CO_2$ concentrations, Limnol. Oceanogr., 46(6), 1378–1391, doi:10.4319/lo.2001.46.6.1378, 2001.

Dortch, Q., Clayton, J. R., Thoresen, S. S. and Ahmed, S. I.: Species differences in accumulation of nitrogen pools in phytoplankton, Mar. Biol., 81(3), 237–250, doi:10.1007/BF00393218, 1984.

Dortch, Q., Clayton, J. R., Thoresen, S. S., Cleveland, J. S., Bressler, S. L. and Ahmed, S. I.: Nitrogen Storage and Use of Biochemical Indexes to Assess Nitrogen Deficiency and Growth-Rate in Natural Plankton Populations, J. Mar. Res., 43(2), 437–464, doi:10.1357/002224085788438621, 1985.

Espie, G. S., Miller, A. G., Canvin, D. T., Espie, G. S., Miller, A. G. and Canvin, D. T.: High Affinity Transport of $CO_2$ in the Cyanobacterium Synechococcus UTEX 625, Plant Physiol., 97, 943–953, 1991.

Flynn, K. J., Fasham, M. J. R. and Hipkin, C. R.: Modelling the interactions between ammonium and nitrate uptake in marine phytoplankton, Philos. Trans. R. Soc. B Biol. Sci., 352(1361), 1625–1645, doi:10.1098/rstb.1997.0145, 1997.

Flynn, K. J., Clark, D. R. and Xue, Y.: Modeling the release of dissolved organic matter by phytoplankton, J. Phycol., 44(5), 1171–1187, doi:10.1111/j.1529-8817.2008.00562.x, 2008.

Fredeen, A. L., Raab, T. K., Rao, I. M. and Terry, N.: Effects of phosphorus nutrition on photosynthesis in Glycine max (L.) Merr., Planta, 181, 399–405, 1990.

Geider, R. J., La Roche, J., Greene, R. M. and Olaizola, M.: Response of the photosynthetic apparatus of Phaeodactylum tricornatum (Bacillariophycea) to nitrate, phosphate, or iron starvation, J. Phycol., 29(6), 755–766, doi:10.1111/j.0022-3646.1993.00755.x, 1993.

Ghyoot, C., Flynn, K. J., Mitra, A., Lancelot, C. and Gypens, N.: Modeling Plankton Mixotrophy: A Mechanistic Model Consistent with the Shuter-Type Biochemical Approach, Front. Ecol. Evol., 5(July), 1–16, doi:10.3389/fevo.2017.00078, 2017.

Giordano, M., Beardall, J. and Raven, J. A.: $CO_2$ Concentrating Mechanisms In Algae: Mechanisms, Environmental Modulation, and Evolution, Annu. Rev. Plant Biol., 56(1), 99–131, doi:10.1146/annurev.arplant.56.032604.144052, 2005.

Goldman, J. C., Mccarthyt, J. J. and Peavy, D. G.: Growth rate influence on the chemical composition of phytoplankton in oceanic waters, Nature, 279(2), 1, doi:10.1038/279210a0, 1979.

Granum, E., Kirkvold, S. and Myklestad, S. M.: Cellular and extracellular production of carbohydrates and amino acids by the marine diatom Skeletonema costatum: Diel variations and effects of N depletion, Mar. Ecol. Prog. Ser., 242(Werner 1977), 83–94, doi:10.3354/meps242083, 2002.

de Groot, C. C., Van Den Boogaard, R., Marcelis, L. F. M., Harbinson, J. and Lambers, H.: Contrasting effects of N and P deprivation on the regulation of photosynthesis in tomato plants in relation to feedback limitation, J. Exp. Bot., 54(389), 1957–1967, doi:10.1093/jxb/erg193, 2003.

Kamalanathan, M., Pierangelini, M., Shearman, L. A., Gleadow, R. and Beardall, J.: Impacts of nitrogen and phosphorus starvation on the physiology of Chlamydomonas reinhardtii, J. Appl. Phycol., 28(3), 1509–1520, doi:10.1007/s10811-015-0726-y, 2016.

Kooijman, S. A. L. M.: The Synthesizing Unit as model for the stoichiometric fusion and branching of metabolic fluxes, Biophys. Chem., 73, 179–188, 1998.

Kooijman, S. A. L. M.: Dynamic Energy Budget theory for metabolic organisation, 3rd ed., Cambridge University Press., 2010.

Lika, K. and Papadakis, I. A.: Modeling the biodegradation of phenolic compounds by microalgae, J. Sea Res., 62(2–3), 135–

146, doi:10.1016/j.seares.2009.02.005, 2009.

Lomas, M. W. and Glibert, P. M.: Comparisons of nitrate uptake, storage, and reduction in marine diatoms and flagellates, J. Phycol., 36, 903–913, 2000.

Lourenço, S. O., Barbarino, E., Lanfer Marquez, U. M. and Aidar, E.: Distribution of intracellular nitrogen in marine microalgae: Basis for calculation of specific nitrogen-to-protein conversion factors, J. Phycol., 34, 798–811, doi:10.1080/0967026032000157156, 1998.

Miller, A. G., Espie, G. S. and Canvin, D. T.: Active Transport of $CO_2$ by the Cyanobacterium *Synechococcus* UTEX 625, Plant Physiol., 87, 677–683, 1988.

Neven, I. A., Stefels, J., van Heuven, S. M. A. C., de Baar, H. J. W. and Elzenga, J. T. M.: High plasticity in inorganic carbon uptake by Southern Ocean phytoplankton in response to ambient $CO_2$, Deep. Res. Part II Top. Stud. Oceanogr., 58(25–26), 2636–2646, doi:10.1016/j.dsr2.2011.03.006, 2011.

Papadakis, I. A., Kotzabasis, K. and Lika, K.: Modeling the dynamic modulation of light energy in photosynthetic algae, J. Theor. Biol., 300, 254–264, doi:10.1016/j.jtbi.2012.01.040, 2012.

Perry, M. J.: Phosphate utilization by an oceanic diatom in phosphorus-limited chemo-stat culture and in the oligotrophic waters of the central North Pacific, Limnol. Oceanogr., 21(January), 88–107, doi:10.4319/lo.1976.21.1.0088, 1976.

Pettersson, K.: Seasonal uptake of carbon and nitrogen and intracellular storage of nitrate in planktonic organisms in the Skagerrak, J. Exp. Mar. Bio. Ecol., 151(1), 121–137, doi:10.1016/0022-0981(91)90020-W, 1991.

Poggiale, J.-C., Baklouti, M., Queguiner, B. and Kooijman, S. a L. M.: How far details are important in ecosystem modelling: the case of multi-limiting nutrients in phytoplankton-zooplankton interactions., Philos. Trans. R. Soc. Lond. B. Biol. Sci., 365(1557), 3495–507, doi:10.1098/rstb.2010.0165, 2010.

Raimbault, P. and Mingazzini, M.: Diurnal variations of intracellular nitrate storage by marine diatoms: effects of nutritional state, J. Exp. Mar. Bio. Ecol., 112(3), 217–232, doi:10.1016/0022-0981(87)90070-0, 1987.

Rost, B., Burkhardt, S., Sültemeyer, D., Riebesell, U., Burkhardt, S. and Sültemeyer, D.: Carbon acquisition of bloom-forming marine phytoplankton, Limnol. Oceanogr., 48(1), 55–67, doi:10.4319/lo.2003.48.1.0055, 2003.

Rotatore, C. and Colman, B.: The active uptake of carbon dioxide by the marine diatoms *Phaeodactytum ticornutum* and *Cyclotella* sp., J. Exp. Bot., 43(4), 571–576, doi:10.1093/jxb/43.4.571, 1992.

Sültemeyer, D., Schmidt, C. and Fock, H. P.: Carbonic anhydrases in higher plants and aquatic microorganisms, Physiol. Plant., 88(1), 179–190, doi:10.1111/j.1399-3054.1993.tb01776.x, 1993.

Talmy, D., Blackford, J., Hardman-Mountford, N. J., Polimene, L., Follows, M. J. and Geider, R. J.: Flexible C:N ratio enhances metabolism of large phytoplankton when resource supply is intermittent, Biogeosciences, 11(4), 5179–5214, doi:10.5194/bgd-11-5179-2014, 2014.

Teira, E., Pazo, M. J., Serret, P. and Fernandez, E.: Dissolved organic carbon production by microbial populations in the Atlantic Ocean, Limnol. Oceanogr., 46(6), 1370–1377, doi:10.4319/lo.2001.46.6.1370, 2001.

Thornton, D. C. O.: Dissolved organic matter (DOM) release by phytoplankton in the contemporary and future ocean , Eur. J. Phycol., 49(1), 20–46, doi:10.1080/09670262.2013.875596, 2014.

Tortell, P. D., Martin, C. L. and Corkum, M. E.: Inorganic carbon uptake and intracellular assimilation by subarctic Pacific phytoplankton assemblages, , 51(5), 2102–2110, 2006.

Urbani, R., Magaletti, E., Sist, P. and Cicero, A. M.: Extracellular carbohydrates released by the marine diatoms Cylindrotheca closterium, Thalassiosira pseudonana and Skeletonema costatum: Effect of P-depletion and growth status, Sci. Total Environ., 353(1–3), 300–306, doi:10.1016/j.scitotenv.2005.09.026, 2005.

---

## Author Comment (AC2) · 17 Dec 2017

Dear Editor,

Please consider below our responses to second referee's comments to our manuscript, entitled "Dissolved organic matter release by phytoplankton in the context of the Dynamic Energy Budget theory", by E. Livanou, A. Lagaria, S. Psarra and K. Lika. We thank referee 2 for the time invested and for the thorough review of our manuscript. In the following, we present our detailed responses to his comments (referee's comments are presented in bold, our replies in blue and relevant changes in the text are highlighted in red to facilitate reading). All citations in our responses are listed at the end of the document. Citations in the original and modified text are included in the references list of the original ms.

**Referee #2**
**This work presents a DEB-based model purporting to describe the production of DOM by phytoplankton under N or P stress. The general topic is important, and does align with the subject areas of the journal.**

**The model is shown to broadly align with a single data set for N-limitation, and is then operated under different conditions with an aim to consider differential consequences of N vs P limitation. I have the following general observations upon this work which I am afraid makes me question the usefulness of the approach and application.**

**Firstly, the structure of the model, as shown in Fig.1, is contrary to that which aligns with the physiology of real phytoplankton. Indeed the structure, with its partitioning of inorganic N and P and within different organic structures appears most strange. I find this very worrying. The model is complex, and claims to be mechanistic (as are DEB models), but it just does not tally.**

**There are also other facets of the model description that are of concern to me: the stoichiometry of cellular components (which is not shown to collectively reproducing observed changes in cellular C:N:P); the explanation of internal ammonium (which in reality is essentially zero) and of internal nitrate (which is also usually very low, and indeed contrary to some reports can never attain a significant % of cell-N because of nitrate-solubility issues) and the interactions between ammonium and nitrate usage (which perpetuate various classic misleading literature articles); the whole rational for DIN and DIP release (perhaps a hangover from the heterotrophic origins of the DEB concept?) appears to be unsupported unless the simulations are running in a light-dark cycle (are they?); comments about cell size variation gloss over the fact that P-limited cells are very much larger than nutrient-replete cells, and N-limited cells are much smaller; and so on.**

**Response**
Regarding the model structure, presented in Fig. 1 in the ms, it follows directly from DEB theory's assumptions that biomass of an organism is partitioned into reserves and structure (Kooijman, 2010). The need to consider multiple reserves follows from DEB theory's strong homeostasis assumption (i.e. the chemical composition of a reserve or a structure does not change during the organism's life cycle), that implies that multiple reserves should be considered, one for each assimilation pathway, when they are independent as is the case for phytoplankton. The partitioning of phytoplankton biomass into different compartments is not new (e.g., Ross and Geider, 2009; Talmy et al., 2014; Ghyoot et al., 2017). Generally, all

elements (e.g. C, N, P) included in the models are partitioned within different compartments (e.g., reserves, structural and functional components of the cells).

Regarding the inorganic reserves the model aligns with existing information about the physiology of phytoplankton cells. As it is also discussed in detail in our response to referee's 1 comment about ammonium and nitrate reserve, a substantial body of literature indicates that phytoplankton cells have the ability to store nitrate and ammonium to measurable quantities (Dortch et al., 1984; Lomas and Glibert, 2000; Lourenço et al., 1998; Raimbault and Mingazzini, 1987). Based on these studies, we include in our model the reserves for nitrate and ammonium in order to simulate the well known behaviour of excess uptake and internal accumulation of inorganic nitrogen when uptake rate exceeds growth rates. This approach is not new as internal pools of inorganic nitrogen (i.e. reserves) are also explicitly modelled in other phytoplankton models (Flynn et al., 1997; Talmy et al., 2014; Ghyoot et al., 2017).

It is acknowledged in the literature that there are significant differences among species in their ability to accumulate large intracellular nitrate or ammonium pools. Our aim was to present a general model for phytoplankton. Differences among species regarding the extent to which ammonium and/or nitrate are accumulating intracellularly can be accommodated by the model by the choice of parameter values. For example, if for a certain species intracellular ammonium pools do not accumulate a very large turnover rate for ammonia reserve would give an extremely low reserve density for ammonia and thus the ammonia reserve could be omitted (Kooijman, 2010).

Regarding inorganic phosphorus reserves, it is well documented that phytoplankton cells store intracellularly excess inorganic phosphorus, as orthophosphate or polyphosphate, not needed immediately to support cell metabolism (Geider and La Roche, 2002; Lin et al., 2016). In addition, inorganic phosphorus reserves have been also described in other modelling studies (John and Flynn, 2000; Ghyoot et al., 2017).

Finally, the concept of the generalized reserves, E, which is assumed to be a mixture of chemical compounds that does not change in composition and thus have a fixed stoichiometry, is used to represent the more complex compounds that can be used as reserves for the phytoplankton, such as lipids, proteins, RNA etc.

In order to clarify the concept of generalised reserves we will modify the text in Sect. 2.1, p.3, lines 31-32 *"[…] to form the generalized reserves (E) which have a fixed stoichiometry."* as follows:

*"[…] to form the generalized reserves (E), which consist of a mixture of chemical compounds (RNA, proteins, lipids etc) and they have a fixed stoichiometry."*

Regarding the interactions among nitrate and ammonium, there are many reports in the literature that suggest that ammonium is taken up preferentially and that assimilation of nitrate is inhibited by a product of ammonium assimilation (e.g. Flynn et al. 1997; Glibert et al. 2015 and references therein). Based on this information, we included the information for ammonium and nitrate interaction in our model.

Regarding the release of DIN and DIP, we would like first to clarify that it stems from the multiple reserves DEB model for phytoplankton as presented in Kooijman (2010), chapter 5. The excretion of inorganic nitrogen and phosphorus from nutrient replete cells has been observed in experimental studies and we provide all the rationale and the literature supporting our choice in the Supplementary Material Sect.S2, p.2, lines 6-35:*"Apart from excretion of DOM […] back to the EP–reserve".*

Apart from that, in our model inorganic nitrogen and phosphorus can be excreted by phytoplankton as the result of maintenance processes and turnover of cellular components. This again stems from the standard DEB model. These nutrients are available for re-assimilation. We recognize that this may be unusual for phytoplankton, thus an alternative would be to redirect regenerated ammonia and phosphorus from maintenance processes back to the corresponding reserves and we intend to add this alternative in the revised manuscript. To answer the referee's question, simulations are not run over light-dark cycle but under constant light intensity.

**Referee #2**
**I therefore have a serious problem with the conceptual basis of the model. The authors have also not explained why their approach has any benefits over any other approach.**

**The model is shown to fit against only one data set, for a diatom, growing under Nlimitation. That data set comes from one of those presented in Flynn et al. 2008, who also conducted a (solely N-based) exploration of the description of DOM release. Very strangely there is no comparison with the utility of this model with that of Flynn et al for N-based scenarios; why have the authors not done this?**

**Response**
Our aim was to present a DEB based model for phytoplankton growth and exudation of DOM thus we did not focus on comparing our modelling approach with that of Flynn et al. (2008). Moreover, our approach captures the two physiological mechanisms of DOM production, without making further assumptions but rather via the existence of two alternative pathways that emerge from the theory. Based on the comments of referee 2 we will include a comparison between the presented model and that of Flynn et al. (2008), in the Discussion under Sect. 4.3, p. 15, line 23 in the revised manuscript. Please see the revised section 4.3, presented as a whole, in our response to the last comment of referee 2 (end of this document).

**Referee #2**
**Further, while this article considers P-limitation (Flynn et al did not) it does so using a framework that is unproven, and hence one that must remain speculative. Indeed, no evidence is presented for how the model handles P-limitation, or indeed how cellular C:N:P varies under N and P limitation. If the DEB model had at least been shown to handle general C-N-P interactions (there are various data sets available to which to conduct such a comparison) then this would not be of such concern.**

**Response**
Elemental composition of biomass (being the sum of reserves and structure) is monitored in terms of Carbon (C), Nitrogen (N) and Phosphorus (P). In the context of DEB theory, biomass

is the sum of reserves and structure. As it is stated in Sect.2.3 of the ms (p.9, lines 28-30), the molecular elemental ratios of biomass are calculated as follows: $n_N = n_{N,E}M_E + M_{E_{NH}} + M_{E_{NO}} + n_{N,V}M_V$ is the total N-mol content, $n_P = n_{P,E}M_E + M_{E_P} + n_{P,V}M_V$ is the total P-mol content and $n_C = n_{C,E}M_E + M_{E_{CH}} + n_{C,V}M_V$ is the total C-mol content of the cells. Thus, $C:N = n_C/n_N$, $C:P = n_C/n_P$, and $N:P = n_N/n_P$. As referee 2 points out, we did not conduct a comparison with data sets for C-N-P limited growth. For our study we were interested in the DOM production by phytoplankton and in order to constrain the model parameters we would need data derived from an axenic phytoplankton culture demonstrating mass balance for all elements considered in the model. However, to our knowledge, this type of data that simultaneously describe carbon, nitrogen and phosphorus dynamics in both the organic and inorganic particulate and dissolved pools are not available for any species. Thus, to cover this gap, in addition to the data set of Flynn et al. (2008), as stated in the methods section (lines 26-27, p.9) the molecular elemental ratios of biomass (i.e. C:P, C:N, N:P) were also used in order to constrain the parameter values so that the resulting ratios (Fig. 8a-c in the ms) would fall close to observable ranges reported in Perry (1976), Goldman et al. (1979) and Flynn et al. (2008). We intend to explicitly state that in the revised manuscript by modifying lines 26-27, p.9 as follows:

*"Furthermore, the molecular elemental ratios of biomass (i.e. C:P, C:N, N:P) were also used in order to constrain the parameter values so that the resulting ratios would fall close to observable ranges reported in Perry (1976), Goldman et al. (1979) and Flynn et al. (2008)."*

In addition, the multiple reserves DEB model for phytoplankton have been used yet again to describe the internal nutrient dynamics of *Prochlorococcus* growing in cultures under balanced, N and P limited growth (Grossowicz et al., 2017). These authors also demonstrated how the DEB model for phytoplankton can handle general C-N-P interactions. Therefore, we do not think that the framework of DEB for phytoplankton models is unproven. In order to dissipate the referee's doubts on the ability of DEB multiple reserves model to handle C-N-P interactions in phytoplankton, we will explicitly cite the relevant papers in the Methods section (Sect. 2.1, p.3, line 17) by modifying the text as follows:

*"The phytoplankton model is based on the DEB multiple reserves model (Kooijman, 2010), that has already been successfully applied to model light, N and P limited growth of phytoplankton (Lorena et al., 2010; Grossowicz et al., 2017)."*

**Referee #2**
**The model is sold for potential deployment in ecosystem simulators but it appears incomplete for such usage, lacking acclimative Chl:C (which is important in DOM modelling as a failure to rapidly modulate C-fixation promotes DOM release) and indeed it has not been shown how the model reacts to light-limitation (which is of importance during bloom development and thence to DOM release).**

**Response**
We suggested that the model could potentially be used in ecosystem simulations in order to resolve the stoichiometry and degradability of DOM produced by phytoplankton, as the model structure allows for a quantification of the production of the distinct size fractions of phytoplankton exudates which can be used for the characterization of their availability for bacteria. Indeed, as referee 2 points out, in our model we did not account for acclimative Chl:C; this could be done for example through light dependence of the parameter $\rho_L$ (eq.6) (Papadakis et al., 2012). Although we agree with referee 2 that this is important for a

potential deployment of our model in ecosystem models, this will require further analysis and data and it is beyond the scope of this ms.

**Referee #2**

**In section 4.2 there is a commentary about P vs N –limitation; I suggest this requires some common basis for reference (perhaps u/Umax?). As it stands the statements appear ambiguous and potentially incorrect.**

**Response**

We based our discussion about P- vs N-limitation on specific growth rates, i.e., growth rate per structural biomass, which is a relative measure. An analytic formula for the ratio of spec. growth rate and the max spec. growth rate (when none of nutrients are limiting) cannot be obtained. Moreover, Fig. 4a in the ms shows that during the first phase of growth, which corresponds to the nutrient-replete phase, indicated also by the minimum C:N and C:P ratios of biomass (Fig. 8a,b in the ms), the specific growth rates are identical. Thus, dividing the specific growth rates by the max. value, observed in the nutrient-replete phase, would result in curves which have the same shape. Therefore, we believe that this measure is not ambiguous and our conclusions are not unjustified.

**Referee #2**

**In section 4.3 is a discussion about forms of DOM. I find the description of DOC, DON, DOP used in this article somewhat confusing; DON and DOP are also components (subsets) of DOC. The discussion lacks a consideration of CNP of DOM forms, and also (again) a comparison with the outputs of the Flynn et al. effort. Just now it is not clear to me what advantages this DEB-based approach may have over any other model. There are also some strange (to me) comments concerning glucose and polysaccharides in this section.**

**Response**

In our model we do not specify the chemical composition of DOM in terms of the various DOM forms (such as amino acids, proteins, nucleic acids etc) but we keep track of the stoichiometry of DOM produced, in terms of C:N:P. DOC consists of carbohydrates, while it can also contain the DOC fraction of organic forms of nitrogen and phosphorus (e.g. proteins, amino acids, P esters etc). The organic forms of nitrogen and phosphorus are referred to as DON and DOP, respectively, which is the standard notation used in many papers and textbooks.

Our approach allows tracking the elemental composition of DOM produced which will be a result of the nutrient availability. Our model shows that, growth under nutrient replete conditions results in DOM production with balanced DOC:DON:DOP ratios as its production is associated mainly with passive leakage. On the other hand, unbalanced growth under nutrient limiting conditions will result in higher rejection fluxes of the non-limiting substrates by the SU and elevated DOC:DON or DOC:DOP ratios (depending on the limiting nutrient).

In order to make model assumptions clearer regarding the relative presence of glucose and heteropolysaccharides of various monemer composition in the excreted DOC, based on both referee's comments, we intend to modify Sect.3.2.3 (p. 13, lines 1-10) *"In order to further investigate […] exuded by the cell due to unbalanced growth (Fogg, 1983; Urbani et al., 2005; Flynn et al., 2008; Borchard and Engel, 2015)"* in the revised ms as follows:

*"Borchard and Engel (2015) showed that in steady-state, P-limited cultures of Emiliania huxleyi glucose was the dominant monomer in both the small size fraction (1–10 kDa) of*

*dissolved polysacchrides and in the particulate fraction (cell content) and less significant in the larger size fractions (>10 kDa), which contained a variety of monomers. They suggested that, due to their size and resemblance to the cellular material, low molecular weight carbohydrates should be released by passive diffusion. On the other hand, high molecular weight carbohydrates, due to their size and distinct composition from the cellular material, should be produced via active exudation (Borchard and Engel, 2015). Furthermore, in cultures of marine diatoms, glucose has been identified as the most abundant monomer during the exponential, nutrient-replete phase of growth in the extracellular carbohydrates. A pronounced decrease of glucose and an increase of heteropolysacharides, containing various monomers, has been observed in the stationary, nutrient-limited phase (Underwood et al., 2004; Urbani et al., 2005). Based on these evidence we define as $DOC_L$ the DOC produced as a result of the fluxes associated with growth ($j_{DOC,G}$ , Fig. 6, dashed line) and death ($j_{DOC,D}$, Fig. 6, dash-dot line) processes and we relate these two fluxes to the mechanism of passive diffusion mechanism. Thus, $DOC_L$ should be small in size and contain mono- and polysacharides, rich in glucose, and also DOC associated with nitrogen or phosphorus containing compounds that can be released from exponentially growing cells or from cell lysis. On the other hand, we define as $DOC_H$ the DOC produced as a result of the $j_{DOC,R}$ flux (Fig. 6, solid line) that corresponds to the rejection flux of unprocessed substrates by the SU. Consequently, this flux is related to the mechanism of active exudation. $DOC_H$ should contain high molecular weight (>10kDa) heteropolysaccharides, poor in glucose, and also DOC associated with nitrogen or phosphorus containing compounds that could be exuded by the cell due to unbalanced growth."*

In addition, in order to make clarify our findings, compare our modelling approach with the Flynn et al. (2008) work and demonstrate the utility of our approach, after comments of both referees, in the revised ms, we intend to replace lines 20-34, p.15 and lines 1-9, p.16 in the Discussion (Sect. 4.3) "In this study, using the DEB model […]consisting of a variety of monomers, have been found to escape bacterial degradation (Obernosterer and Herndl, 1995; Hama and Yanagi, 2001; Puddu et al., 2003)" with:

*"In this study, using the DEB model for phytoplankton (Kooijman, 2010), we were able to discriminate between the two major conceptual mechanisms of DOM release, i.e., passive diffusion and active exudation, in contrast to existing phytoplankton models that involve DOM exudation (Van Den Meersche et al., 2004; Schartau et al., 2007; Flynn et al., 2008; Kreus et al., 2014) but do not discriminate between the mechanisms of DOM release. For example, the most complex model presented in Flynn et al. (2008) employed an empirical description that related the relative rate of leakage of DOC and DON to the N:C status of the cells. On the contrary, in our modeling approach, the theory of SU quantifies the active exudation of the non-limiting compounds. Since the nutrients are taken up independently, one or more catabolic fluxes can limit the synthesis of the generalized reserves $E'$.  In that case, the non limiting "molecules" will occupy the binding sites of the $SU_2$ but they will not be processed further due to the absence of the limiting flux and, thus, they will be rejected by the $SU_2$. Subsequently, a fraction of this rejection flux will be excreted.  In that way, the effect of nutrient limitation on exudation rate is accounted for. Furthermore, Flynn et al. (2008) assumed a higher rate of leakage until the external concentration attained a critical value in order to account for the rapid accumulation of DOM during the initial stages of the culture. On the other hand, in our model, based on DEB theory's assumptions for product formation, we describe a second process of DOM excretion which is stoichiometrically coupled to the growth rate and results in high rates of DOM production during the initial nutrient-replete phase of the culture. This can be seen as an overhead for growth as this material is passively leaked outside the cell. Thus, the advantage of our approach lies in its ability to capture the*

*two physiological mechanisms of DOM production. It does so without making further assumptions but rather via the existence of two alternative pathways that emerge from the theory.*

*Based on experimental evidence we assumed that the different processes contributing to DOC release produce two distinct types of DOC (i.e., $DOC_L$, $DOC_H$). $DOC_L$, which is related to growth and death processes and thus to the passive diffusion mechanism, is expected to contain low molecular weight carbohydrates that have a similar composition as the cellular material with high content of glucose (Borchard and Engel, 2015), while $DOC_H$ which is related to the rejection flux of unprocessed substrates by the SU and the active exudation mechanism, has a more distinct composition from the cellular material and is rich in high molecular weight heteropolysaccharides, consisting of a variety of monomers (Biersmith and Benner, 1998; Borchard and Engel, 2015). As such, the model suggests that the relative importance of the two mechanisms and, thus, the relative presence of high and low molecular weight carbohydrates with different molecular composition signatures, is dependent on the nutrient status of the cells. Our approach is different of that of Flynn et al (2008) since they did not distinguish between the two mechanisms of DOM production. Thus, in their model they take into account only leakage, which is related to the nutrient status of the cells and produce low molecular weight DOC such as mono and disaccharides and DOC associated with amino acids. High molecular weight DOC is associated with proteins and nucleic acids and it is produced only via cell lysis, that is enhanced under suboptimal growth conditions, while cell lysis will also result in the leakage of low molecular weight DOM stored intracellularly (Flynn et al., 2008).*

*The molecular composition of DOC released may have implications for its subsequent utilization by bacteria. Many studies have shown that exudates, rich in glucose, are taken up rapidly by bacteria while heteropolysaccharides, consisting of a variety of monomers, have been found to escape bacterial degradation (Obernosterer and Herndl, 1995; Hama and Yanagi, 2001; Puddu et al., 2003). Thus, the novelty of our model is that it allows the quantification of the production fluxes associated with the two classes of DOC ($DOC_L$, $DOC_H$) that will contain carbohydrates with different molecular composition signatures and thus, different degree of degradability by bacteria. Furthermore, our model setup allows for the tracking of the elemental composition of photosynthetically produced DOM, which is also important information for the degradability of DOM.''*

**There are various detailed comments that I could give to help the authors, but just now I think that I need to see:**
**i) a more acceptable conceptual basis (I do not believe that Fig.1 does this),**
**ii) a demonstration that the whole model can describe dynamic C:N:P experimental data series,**
**iii) a more rigorous set of comparisons with Flynn et al (whose data they use, but then for some reason never further discuss in comparative terms even in the context of Nlimited growth) and indeed with the functionality of other models (ERSEM springs to mind).**

We hope that our responses cover referee's points i)-iii) and that they will dissipate his doubts on the usefulness and application of our modelling approach.

**References**

Dortch, Q., Clayton, J. R., Thoresen, S. S. and Ahmed, S. I.: Species differences in accumulation of nitrogen pools in phytoplankton, Mar. Biol., 81(3), 237–250, doi:10.1007/BF00393218, 1984.

Flynn, K. J., Fasham, M. J. R. and Hipkin, C. R.: Modelling the interactions between ammonium and nitrate uptake in marine phytoplankton, Philos. Trans. R. Soc. B Biol. Sci., 352(1361), 1625–1645, doi:10.1098/rstb.1997.0145, 1997.

Flynn, K. J., Clark, D. R. and Xue, Y.: Modeling the release of dissolved organic matter by phytoplankton, J. Phycol., 44(5), 1171–1187, doi:10.1111/j.1529-8817.2008.00562.x, 2008.

Geider, R. and La Roche, J.: Redfield revisited: variability of C:N:P in marine microalgae and its biochemical basis Redfield revisited: variability of C:N:P in marine microalgae and its biochemical basis, Eur. J. Phycol., 37(September 2012), 37–41, doi:10.1017/S0967026201003456, 2002.

Ghyoot, C., Flynn, K. J., Mitra, A., Lancelot, C. and Gypens, N.: Modeling Plankton Mixotrophy: A Mechanistic Model Consistent with the Shuter-Type Biochemical Approach, Front. Ecol. Evol., 5(July), 1–16, doi:10.3389/fevo.2017.00078, 2017.

Glibert, P. M., Wilkerson, F. P., Dugdale, R. C., Raven, J. A., Dupont, C. L., Leavitt, P. R., Parker, A. E., Burkholder, J. M. and Kana, T. M.: Pluses and minuses of ammonium and nitrate uptake and assimilation by phytoplankton and implications for productivity and community composition, with emphasis on nitrogen-enriched conditions, Limnol. Oceanogr., 165–197, doi:10.1002/lno.10203, 2015.

Goldman, J. C., Mccarthyt, J. J. and Peavy, D. G.: Growth rate influence on the chemical composition of phytoplankton in oceanic waters, Nature, 279(2), 1, doi:10.1038/279210a0, 1979.

Grossowicz, M., Marques, G. M. and van Voorn, G. A. K.: A dynamic energy budget (DEB) model to describe population dynamics of the marine cyanobacterium Prochlorococcus marinus, Ecol. Modell., 359, 320–332, doi:10.1016/j.ecolmodel.2017.06.011, 2017.

John, E. H. and Flynn, K. J.: Modelling phosphate transport and assimilation in microalgae; how much complexity is warranted?, Ecol. Modell., 125(2–3), 145–157, doi:10.1016/S0304-3800(99)00178-7, 2000.

Kooijman, S. A. L. M.: Dynamic Energy Budget theory for metabolic organisation, 3rd ed., Cambridge University Press., 2010.

Lin, S., Litaker, R. W. and Sunda, W. G.: Phosphorus physiological ecology and molecular mechanisms in marine phytoplankton, J. Phycol., 1, 10–36, doi:10.1017/CBO9781107415324.004, 2016.

Lomas, M. W. and Glibert, P. M.: Comparisons of nitrate uptake, storage, and reduction in marine diatoms and flagellates, J. Phycol., 36, 903–913, 2000.

Lourenço, S. O., Barbarino, E., Lanfer Marquez, U. M. and Aidar, E.: Distribution of intracellular nitrogen in marine microalgae: Basis for calculation of specific nitrogen-to-protein conversion factors, J. Phycol., 34, 798–811, doi:10.1080/0967026032000157156, 1998.

Papadakis, I. A., Kotzabasis, K. and Lika, K.: Modeling the dynamic modulation of light energy in photosynthetic algae, J. Theor. Biol., 300, 254–264, doi:10.1016/j.jtbi.2012.01.040, 2012.

Perry, M. J.: Phosphate utilization by an oceanic diatom in phosphorus-limited chemo-stat culture and in the oligotrophic waters of the central North Pacific, Limnol. Oceanogr., 21(January), 88–107, doi:10.4319/lo.1976.21.1.0088, 1976.

Raimbault, P. and Mingazzini, M.: Diurnal variations of intracellular nitrate storage by marine diatoms: effects of nutritional state, J. Exp. Mar. Bio. Ecol., 112(3), 217–232, doi:10.1016/0022-0981(87)90070-0, 1987.

Ross, O. N. and Geider, R. J.: New cell-based model of photosynthesis and photo-acclimation: accumulation and mobilisation of energy reserves in phytoplankton, Mar. Ecol. Prog. Ser., 383, 53–71, doi:10.3354/meps07961, 2009.

Talmy, D., Blackford, J., Hardman-Mountford, N. J., Polimene, L., Follows, M. J. and Geider, R. J.: Flexible C:N ratio enhances metabolism of large phytoplankton when resource supply is intermittent, Biogeosciences, 11(4), 5179–5214, doi:10.5194/bgd-11-5179-2014, 2014.

---

## Referee Comment (RC3) · Anonymous Referee #3 · 18 Dec 2017

**General comments**

The authors propose a phytoplankton dynamics model based on DEB theory. They focus the application of their model for explaining the fluxes of DOM release by phytoplankton cells, by considering passive diffusion associated to cell lysis and active exudation associated to unprocessed substrates under stoichiometric constraints. Even if I'm convinced that DEB theory may provide mechanistic models which could help us to understand the processes studied here, I'm have several major problems with the present version of the paper. The first one concerns the model validation, the second one is about some choices in the processes formulation, and the third one concerns

the model presentation.

**Specific comments**

- The authors use data published in Flynn et al., 2008 for comparing model outputs to experimental results. There is no description of the methods used : which distance between data and model is used, how is it optimized, which algorithm is used for the ODE's? Moreover, there is a large number of parameters and there is no information on their correlation in the estimation procedure, the set of data is rather reduced with respect to the number of parameters. Could the authors give some uncertainties on their model output?

- Most of the fluxes formulations are based on SU's dynamics, but the schemes are not provided (maybe they could be the object of a supplementary material). However, among all these formulations, two of them have not the SU's support. This is the case of formulas (5) and (6). These formulations are far from standard ones and are not explained. And furthermore, the square on the phosphorus content $q_P$ and the nitrogen content $q_N$ in the reserves is a surprising refinement, what is the model sensitivity to such a refinement? Could the authors show some data to validate these formulations? Another point is the fact that in the subsection dealing with the growth rate, the authors omit to mention that the growth rate is not explicit. I'm not sure that this can be understood easily by readers who did not try to use multi-reserve DEB models before. This is probably not a strong constraint for the present paper, but having to solve an algebraic equation to calculate the phytoplankton growth rate at each time step could be a strong limitation for 3D-biogeochemical models at large scales. This could at least been mentioned and if the authors have suggestions to solve this, it could be an interesting contribution.

- The model description should be accompanied by a scheme more adapted than Figure 1, I mean that a scheme focused on the fluxes included in the model would

help to follow the model description. The subsection 2.2 should have another name, like "Formulation of processes" instead of "Model equations" because the model equations are already provided in the previous subsection, with undefined process formulations. As it is done, the model description is quite hard to follow.

Since the results and discussions could strongly depend on arbitrary choices (formulations) and parameter values (estimations), it is hard to be convinced about their validity or about their degree of generality now.